# Beta-catenin signaling regulates barrier-specific gene expression in circumventricular organ and ocular vasculatures

Yanshu Wang[1,2], Mark F Sabbagh[1,3], Xiaowu Gu[1,2], Amir Rattner[1], John Williams[1,2], Jeremy Nathans[1,2,3,4]*

[1]Department of Molecular Biology and Genetics, Johns Hopkins University School of Medicine, Baltimore, United States; [2]Howard Hughes Medical Institute, Johns Hopkins University School of Medicine, Baltimore, United States; [3]Department of Neuroscience, Johns Hopkins University School of Medicine, Baltimore, United States; [4]Department of Ophthalmology, Johns Hopkins University School of Medicine, Baltimore, United States

**Abstract** The brain, spinal cord, and retina are supplied by capillaries that do not permit free diffusion of molecules between serum and parenchyma, a property that defines the blood-brain and blood-retina barriers. Exceptions to this pattern are found in circumventricular organs (CVOs), small midline brain structures that are supplied by high permeability capillaries. In the eye and brain, high permeability capillaries are also present in the choriocapillaris, which supplies the retinal pigment epithelium and photoreceptors, and the ciliary body and choroid plexus, the sources of aqueous humor and cerebrospinal fluid, respectively. We show here that (1) endothelial cells in these high permeability vascular systems have very low beta-catenin signaling compared to barrier-competent endothelial cells, and (2) elevating beta-catenin signaling leads to a partial conversion of permeable endothelial cells to a barrier-type state. In one CVO, the area postrema, high permeability is maintained, in part, by local production of Wnt inhibitory factor-1.
DOI: https://doi.org/10.7554/eLife.43257.001

*For correspondence:
jnathans@jhmi.edu

## Introduction

Throughout the body, capillaries exhibit tissue- and organ-specific specializations of structure and function (*Aird, 2007a*; *Aird, 2007b*; *Potente and Mäkinen, 2017*). In the CNS, the blood-brain barrier (BBB) and blood-retina barrier (BRB) are characterized by numerous specializations in vascular endothelial cell (EC) structure and function, including elaboration of tight junctions, suppression of transcytotic pathways, and production of small molecule transporters and extrusion pumps (*Daneman and Prat, 2015*; *Zhao et al., 2015*). [See *Table 1* for a list of abbreviations used in this paper.] The BBB/BRB program of EC differentiation is controlled, at least in part, by beta-catenin signaling (canonical Wnt signaling) that is activated by ligands Wnt7a, Wnt7b, and Norrin. These ligands are produced by glia and neurons, and they activate receptors and co-receptors on the surface of ECs (*Liebner et al., 2008*; *Stenman et al., 2008*; *Daneman et al., 2009*; *Wang et al., 2012*; *Zhou and Nathans, 2014*; *Zhou et al., 2014*; *Posokhova et al., 2015*). Signals from pericytes are also important, as pericyte loss leads to a loss of barrier integrity (*Armulik et al., 2010*; *Daneman et al., 2010*).

Gain-of-function and loss-of-function experiments in mice have demonstrated an ongoing requirement for beta-catenin signaling in CNS ECs to maintain the barrier state (*Liebner et al., 2008*;

**Table 1.** Abbreviations.

| | |
|---|---|
| A | anterior |
| AP | area postrema or anterior pituitary |
| ATAC | assay for transposase-accessible chromatin |
| BBB | blood-brain barrier |
| BRB | blood-retina barrier |
| CB | ciliary body |
| CC | choriocapillaris |
| CNS | central nervous system |
| CP | choroid plexus |
| CSF | cerebrospinal fluid |
| CVO | circumventricular organ |
| EC | vascular endothelial cell |
| FACS | fluorescent-activated cell sorting |
| IP | intraperitoneal |
| ME | median eminence |
| PLVAP | plasmalemma vesicle associated protein |
| PP | posterior pituitary |
| P | posterior |
| RPE | retinal pigment epithelium |
| SFO | subfornical organ |
| TEM | transmission electron microscopy |
| VOLT | vascular organ of the lamina terminalis |
| WIF1 | Wnt inhibitory factor-1 |
| WT | wild type |
| 4HT | 4-hydroxytamoxifen |

DOI: https://doi.org/10.7554/eLife.43257.002

*Wang et al., 2012*). In the mature CNS vasculature, acute loss of Frizzled4 (Fz4), one of the EC receptors for beta-catenin signaling, leads to a cell-autonomous loss of barrier integrity in the retina, cerebellum, and olfactory bulb. Conversely, acute induction of Norrin production in the CNS of an adult *Ndp* null mouse [*Norrie disease protein* (*Ndp*) is the gene coding for Norrin] restores BBB integrity in the cerebellum. Intriguingly, constitutive activation of beta-catenin signaling in ECs in the choroid plexus, the site of cerebrospinal fluid (CSF) production, leads to the cell autonomous expression of a BBB marker [the tight junction protein Claudin5 (CLDN5)] and repression of a marker of non-BBB vasculature [plasmalemma vesicle-associated protein (PLVAP), a structural component of fenestrae (*Stan et al., 2004*; *Zhou et al., 2014*)]. These experiments indicate that the BBB state remains plastic throughout life and that beta-catenin signaling in CNS ECs is required on a continuous basis for maintenance of the BBB state.

The mammalian brain contains a set of small midline structures, the circumventricular organs (CVOs), that are unusual in having a dense capillary plexus composed of ECs with high permeability (*Figure 1A*; *Gross, 1992*; *Duvernoy and Risold, 2007*; *Kaur and Ling, 2017*). CVOs can be divided into two categories, sensory and secretory. The vascular organ of the lamina terminalis (VOLT), the subfornical organ (SFO), and the area postrema comprise the sensory CVOs. The VOLT and SFO monitor serum osmolarity and together regulate blood volume, blood pressure, and electrolyte balance by controlling vasopressin release, which, in turn, controls thirst and renal salt and water retention (*Toney et al., 2003*; *Hiyama et al., 2004*; *Oka et al., 2015*; *Hiyama and Noda, 2016*). The SFO additionally monitors glucose and other small molecule nutrients and regulates energy homeostasis (*Fry and Ferguson, 2007*; *Medeiros et al., 2012*). The area postrema monitors serum composition and controls an emetic response to toxic compounds (*Borison, 1989*). The posterior pituitary,

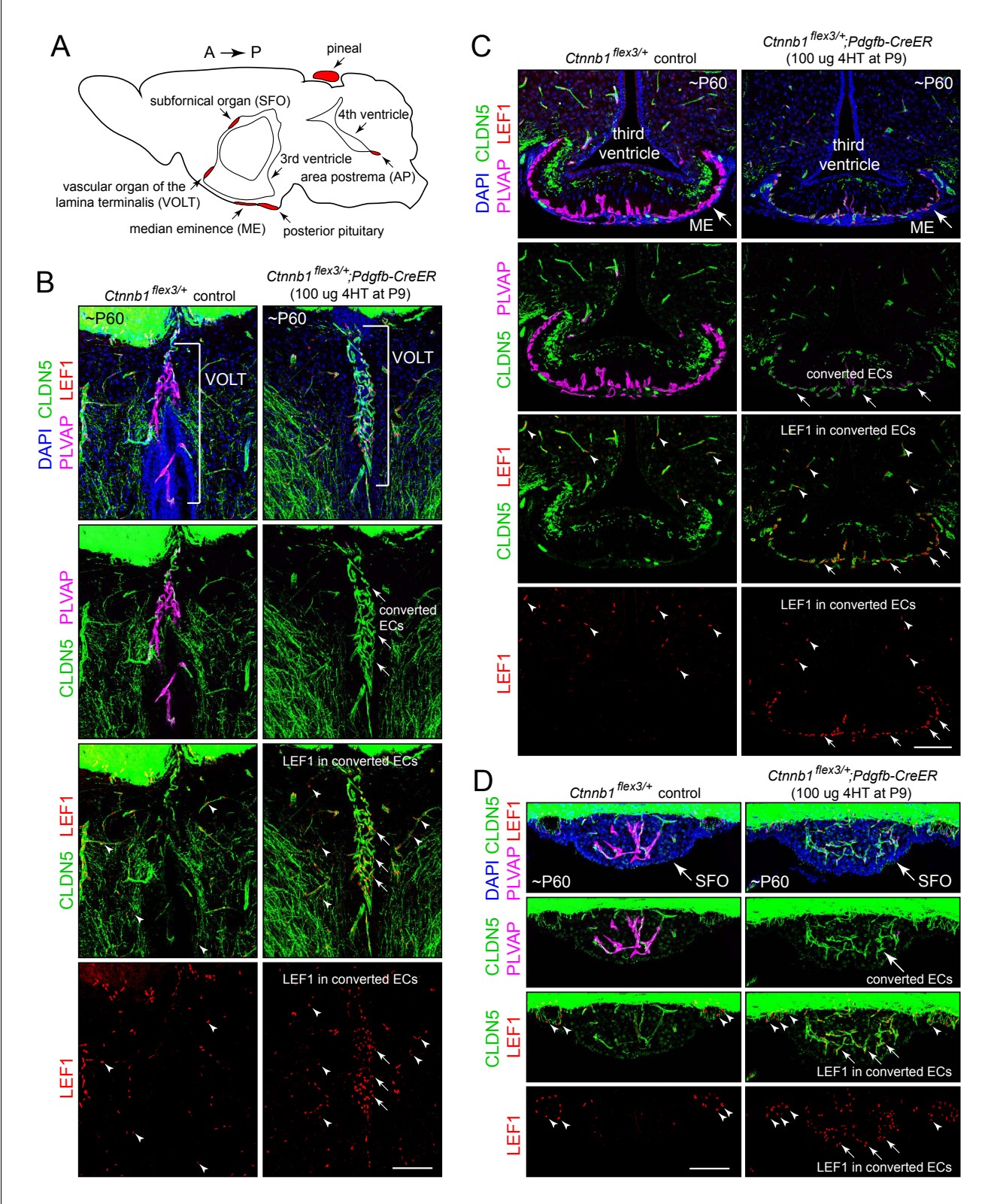

**Figure 1.** Stabilizing beta-catenin in ECs switches the ME, SFO, and VOLT vasculatures to a BBB-like state. (**A**) Schematic of a midline sagittal section of the mouse brain indicating the locations of the CVOs (red). A, anterior; P, posterior. (**B–D**) Coronal sections showing the VOLT (**B**), ME (**C**), and SFO (**D**) from a control mouse (left) and from a mouse with EC-specific deletion of *Ctnnb1* exon 3 (right), immunostained as indicated. In control brains (left panels), ECs in the VOLT, ME, and SFO are CLDN5-/PLVAP+/LEF1-. In mice with EC-specific deletion of *Ctnnb1* exon 3 (right panels), ECs in the VOLT,

*Figure 1 continued on next page*

*Figure 1 continued*

ME, and SFO are CLDN5+/PLVAP-/LEF1+ ('converted ECs'). In both genotypes, ECs in the surrounding brain parenchyma are CLDN5+/PLVAP-/LEF1+. In the bottom 2 rows of images, arrows point to LEF1 in EC nuclei within the VOLT, ME, and SFO with deletion of *Ctnnb1* exon 3. Arrowheads point to LEF1 in EC nuclei in the adjacent brain parenchyma, which are unaffected by deletion of *Ctnnb1* exon 3. The efficiency of CreER-mediated recombination in ECs is ~80–90%, as judged by the conversion of EC markers. In this and all other figures, the age of the mouse in postnatal days (P) is embedded in the image panels, and the dose of 4-hydroxytamoxifen (4HT) and age at its delivery is indicated next to the genotype. Unless noted otherwise, tissues were fixed by cardiac perfusion of paraformaldehyde in physiologic buffer. Scale bars, 100 μm.

DOI: https://doi.org/10.7554/eLife.43257.003

The following figure supplements are available for figure 1:

**Figure supplement 1.** Stabilizing beta-catenin in ECs switches the SFO and pineal vasculatures to a BBB-like state.

DOI: https://doi.org/10.7554/eLife.43257.004

**Figure supplement 2.** Nuclear LEF1 accumulates in phenotypically WT *Fz4*+/- retinal ECs and is undetectable in *Fz4*-/- retinal ECs.

DOI: https://doi.org/10.7554/eLife.43257.005

the median eminence (ME), and the pineal comprise the secretory CVOs and are part of the neuro-endocrine system. The posterior pituitary releases oxytocin and vasopressin into the systemic circulation, and the ME is the site of secretion of hypothalamic hormones into the hypophyseal-portal system to control release of the corresponding target hormones from the anterior pituitary (*Guillemin, 1978*; *Schally, 1978*; *Amar and Weiss, 2003*). The pineal releases melatonin to control circadian behavior (*Axelrod, 1974*).

Like the CVOs, the choroid plexus is composed of ECs with high permeability. Although contained within the CNS, the choroid plexus is not generally considered a CVO because the choroid plexus capillaries are not in direct contact with neurons or glia but are instead surrounded by a specialized epithelium, all of which is surrounded by CSF (*Johanson et al., 2011*). Two regions of the ocular vasculature also contain ECs with high permeability: the choriocapillaris and the ciliary body (*Alm, 1992*). The choriocapillaris is a flattened monolayer of capillaries adjacent to the retinal pigment epithelium (RPE), and it serves as the vascular supply for both the RPE and the photoreceptors, with the RPE monolayer providing the outer blood-retina barrier (BRB). The ciliary body, at the anterior edge of the retina, is the source of aqueous humor, and is therefore analogous in its function to the choroid plexus. The extent to which these specialized vascular beds in the brain and eye (the CVOs, choroid plexus, ciliary body, and choriocapillaris) share molecular properties is largely unexplored.

The foregoing summary suggests the possibility that the distinction between the barrier-type vasculature that serves most of the CNS and the specialized non-barrier vasculatures in the eye and brain could be due, at least in part, to differences in beta-catenin signaling in their respective ECs. The experiments described here support this idea. In particular, we observe that ECs in the CVO vasculature convert to a BBB-like state upon induction of beta-catenin signaling, as judged by changes in (1) ultrastructure and protein composition, (2) expression of BBB vs. non-BBB genes, and (3) chromatin accessibility near BBB genes. We also find that loss of Wnt inhibitory factor-1 (WIF1), a secreted Wnt binding protein that reduces beta-catenin signaling (*Hsieh et al., 1999*) and is expressed specifically in the area postrema, leads to a partial conversion of area postrema ECs to a BBB-like state.

## Results

### Reduced beta-catenin signaling in CVO ECs

The CVO and choroid plexus vasculatures are readily visualized in the wild type (WT) mouse brain by immunostaining for PLVAP, which is abundant in the VOLT, ME, SFO, area postrema, pineal, anterior pituitary, posterior pituitary, and choroid plexus vasculatures but absent from the adjacent BBB-type CNS vasculature (left panels in each panel set in *Figures 1*, *2* and *3A–D"*, and *Figure 1—figure supplement 1*). The BBB markers CLDN5 and the glucose transporter GLUT1 show the reciprocal pattern of accumulation. In our analyses, we have included the anterior pituitary, a peripheral neuroendocrine organ, as a point of comparison with the posterior pituitary and other CVOs.

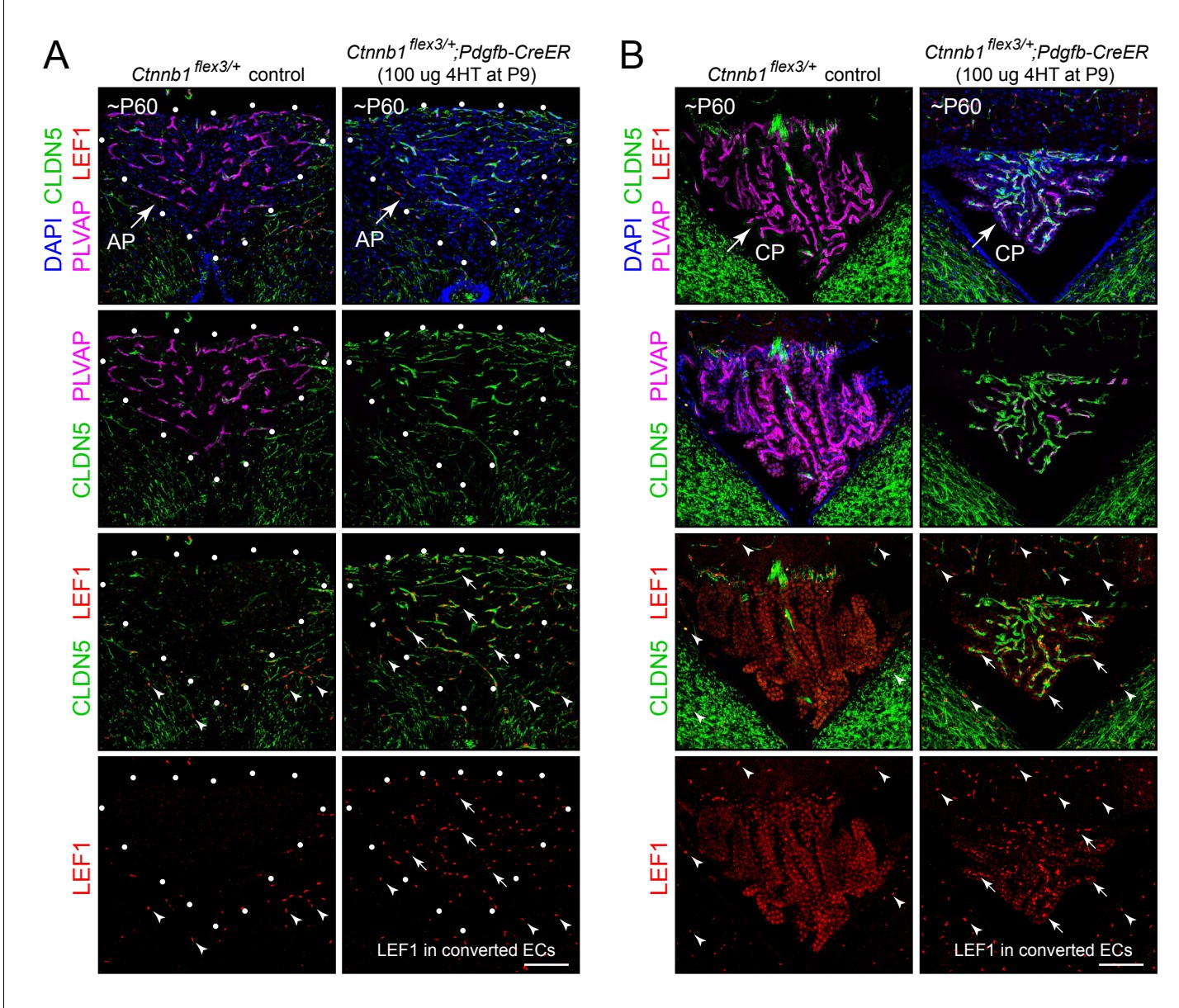

**Figure 2.** Stabilizing beta-catenin in ECs switches the area postrema and choroid plexus vasculatures to a BBB-like state. (**A,B**) Coronal sections showing the area postrema (A; demarcated by white dots and abbreviated AP) and choroid plexus (B; abbreviated CP) from a control mouse (left panels) and from a mouse with EC-specific deletion of *Ctnnb1* exon 3 (right panels), immunostained as indicated. In the area postrema and choroid plexus, ECs in the control are CLDN5-/PLVAP+/LEF1-, and ECs with deletion of *Ctnnb1* exon 3 are CLDN5+/PLVAP-/LEF1+. ECs in the surrounding brain parenchyma are CLDN5+/PLVAP-/LEF1+. In the right panels of the bottom 2 rows of images, arrows point to LEF1 in EC nuclei within the area postrema and choroid plexus with deletion of *Ctnnb1* exon 3. Arrowheads in the bottom 2 rows of images point to LEF1 in EC nuclei in the adjacent brain parenchyma, which are unaffected by deletion of *Ctnnb1* exon 3. The efficiency of CreER-mediated recombination in ECs is >90% in the area postrema and ~80–90% in the choroid plexus, as judged by the conversion of EC markers. Scale bars, 100 μm.

DOI: https://doi.org/10.7554/eLife.43257.006

The following figure supplements are available for figure 2:

**Figure supplement 1.** Stabilizing beta-catenin in ECs induces expression of MDR1 in area postrema ECs.
DOI: https://doi.org/10.7554/eLife.43257.007
**Figure supplement 2.** Stabilizing beta-catenin in ECs induces expression of ZO1 in SFO and choroid plexus ECs.
DOI: https://doi.org/10.7554/eLife.43257.008
**Figure supplement 3.** Stabilizing beta-catenin in ECs induces expression of the canonical Wnt reporter *R26-8xTCF/LEF-LSL-H2B-GFP* and accumulation of GLUT1 in ECs in the area postrema.

*Figure 2 continued on next page*

*Figure 2 continued*

DOI: https://doi.org/10.7554/eLife.43257.009

**Figure supplement 4.** Assessing the specificity and sensitivity of the *R26-8xTCF/LEF-LSL-H2B-GFP* canonical Wnt reporter in the retinal vasculature by assessing its response to decreased or increased beta-catenin signaling.

DOI: https://doi.org/10.7554/eLife.43257.010

LEF1, a member of the TCF/LEF transcription factor family that binds to beta-catenin to mediate the canonical Wnt response (*Eastman and Grosschedl, 1999*), accumulates in some cell types, including CNS ECs, in response to beta-catenin signaling (*Planutiene et al., 2011*). This phenomenon is demonstrated in *Figure 1—figure supplement 2*, where *Fz4$^{CKO/-}$;Pdgfb-CreER* mice that are mosaic for EC-specific loss of *Fz4* show nuclear accumulation of LEF1 in phenotypically WT *Fz4$^{+/-}$* retinal ECs (CLDN5+/PLVAP- cells) but show no detectable LEF1 in adjacent *Fz4$^{-/-}$* retinal ECs (CLDN5-/PLVAP+ cells). To assess the level of beta-catenin signaling in the CVO and choroid plexus vasculatures in the WT adult mouse brain, we visualized CLDN5, PLVAP, and LEF1 in the CVO and choroid plexus vasculatures (left panels in each panel set in *Figures 1, 2* and *3A–D"*). In the VOLT, ME, SFO, area postrema, and choroid plexus, the non-BBB (CLDN5-/PLVAP+) ECs lack detectable LEF1, whereas in the surrounding CNS tissue BBB+ (CLDN5+/PLVAP-) ECs show strong nuclear accumulation of LEF1 (arrowheads in the left panels in each panel set in *Figures 1* and *2*). Similarly, the WT pineal and posterior pituitary lack detectable nuclear LEF1 accumulation (left panels of *Figure 1—figure supplement 1B* and *Figure 3C' and C"*). This differential distribution of LEF1 suggests that beta-catenin signaling is high in BBB+ ECs and low in CVO and choroid plexus ECs.

## The effect of activating beta-catenin signaling in CVO ECs

We next asked whether expressing a stabilized derivative of beta-catenin specifically in ECs – and thereby artificially activating beta-catenin signaling – could shift the properties of the CVO vasculature to a more BBB-like state. In these experiments, exon 3 of the beta-catenin gene (*Ctnnb1*), which encompasses the sites of phosphorylation that lead to beta-catenin degradation, was excised from one *Ctnnb1* allele by Cre-mediated recombination in CNS ECs using a *Pdgfb-CreER* transgene (*Harada et al., 1999*); the allele with *loxP* sites flanking exon three is referred to as *Ctnnb1$^{flex3}$*. As seen in the right panels in each panel set in *Figures 1, 2* and *3A–D"* and *Figure 1—figure supplement 1*, for ECs in each of the six CVOs and in the choroid plexus, beta-catenin stabilization leads to the loss of PLVAP and the accumulation of plasma membrane CLDN5 and nuclear LEF1 (arrows in *Figures 1* and *2* point to LEF1+ EC nuclei). In the posterior pituitary shown in *Figure 3*, excision of *Ctnnb1* exon three occurred in ~80% of ECs, permitting a side-by-side comparison of individual ECs with or without constitutively activated beta-catenin signaling. Among ECs, CLDN5, GLUT1, and LEF1 are co-expressed and are anti-correlated with expression of PLVAP (*Figures 1–3* and *Figure 1—figure supplement 1*; GLUT1 immunostaining is shown for only a subset of the CVOs). In those CVO ECs that exhibit a phenotypic response to beta-catenin stabilization, the change in gene expression appears to occur in an all-or-none fashion, converting a CLDN5-/PLVAP+/LEF1- state to a CLDN5+/PLVAP-/LEF+ state, with very few cells showing partial conversion.

To extend the analysis of the BBB vs. non-BBB phenotype, we examined the distribution of the multi-drug resistance pump MDR1 and the tight junction proteins Occludin (OCLN) and Zonula Occludens-1 (ZO1), which are markers for BBB+ ECs. [As a technical point, we observed little immunoreactivity for these three proteins in paraformaldehyde (PFA) fixed tissues, but found that immunoreactivity was preserved with cardiac perfusion of phosphate buffered saline followed by methanol, a protocol that permits vibratome sectioning of the brain following its rehydration but also likely compromises preservation of subcellular structure relative to PFA fixation.] *Figure 2—figure supplements 1* and *2* show, respectively, sagittal sections of the area postrema and of the SFO and adjacent choroid plexus in mice with mosaic excision of *Ctnnb1* exon 3 in ~90% of ECs. MDR1 and ZO1 are induced in *Ctnnb1* exon 3-deleted ECs in each of these structures (MDR1 is shown for the area postrema and ZO1 is shown for the SFO), and the distribution of MDR1+ and ZO1+ ECs are anti-correlated with the distribution of PLVAP+ ECs, which marks the ECs that failed to excise *Ctnnb1* exon 3. Importantly, the subcellular distribution of ZO1 is consistent with its accumulation at junctional complexes (*Figure 2—figure supplement 2B and B'*, and enlargements b and b'). Unlike CLDN5, ZO1 is not uniformly induced across CVOs; for example, it is only minimally induced in the

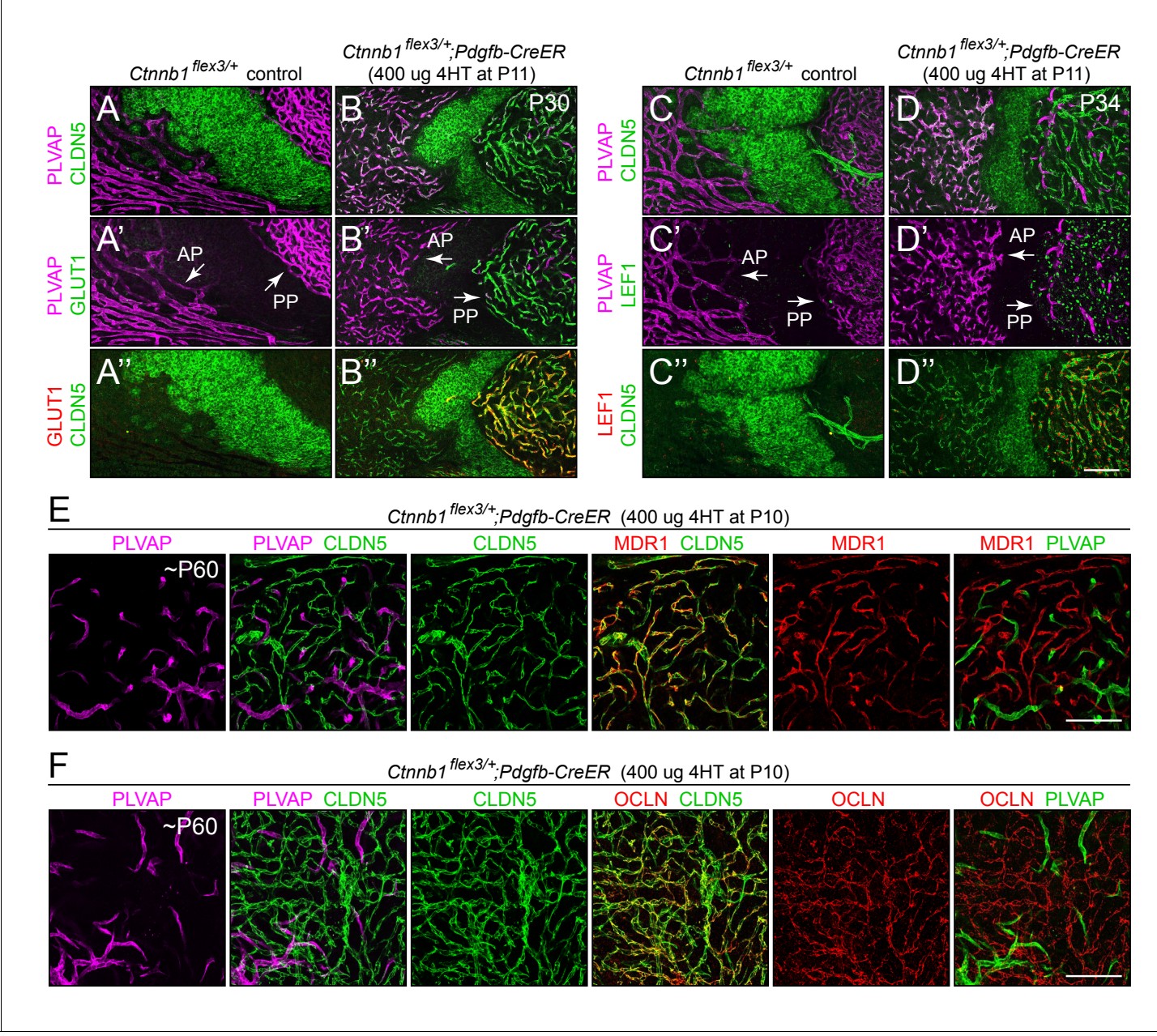

**Figure 3.** Stabilizing beta-catenin in ECs switches posterior pituitary ECs more than anterior pituitary ECs to a BBB-like state. (A–D") Regions of whole-mount pituitaries spanning the junction of the anterior lobe (left), inter-lobe region (middle), and posterior lobe (right) from two control mice (left set of panels) and from two mice with EC-specific deletion of *Ctnnb1* exon 3 (right set of panels), immunostained as indicated. With EC-specific deletion of *Ctnnb1* exon 3, ECs in the posterior pituitary become CLDN5+/PLVAP-/GLUT1+/LEF1+, whereas anterior pituitary ECs show low levels of CLDN5, a modest reduction in the level of PLVAP, a level of GLUT1 that is below the limit of detection, and rare LEF1+ ECs. In the genetically mosaic *Ctnnb1*flex3/+*;Pdgfb-CreER* posterior pituitary vasculature, PLVAP is anti-correlated with CLDN5, LEF1, and GLUT1 on a cell-by-cell basis (**B–B"** and **D–D"**). GLUT1 and LEF1 are shown with different colors in B' and D' to highlight their co-localization with CLDN5 and their anti-correlation with PLVAP in the posterior pituitary. (**E,F**) As in A-D', except images are entirely within the posterior lobe and the pituitaries were immunostained for PLVAP, CLDN5, and either MDR1 or OCLN. In the right-most panels, PLVAP is false-colored green for ease of comparison. The efficiency of CreER-mediated recombination in ECs is ~80%, as judged by the conversion of EC markers. AP, anterior pituitary. PP, posterior pituitary. Scale bar for A-F, 100 μm.
DOI: https://doi.org/10.7554/eLife.43257.011

The following figure supplements are available for figure 3:

**Figure supplement 1.** CLDN5 and OCLN co-localize to cell-cell junctions in posterior pituitary ECs that have been converted to a BBB-like state by beta-catenin stabilization.
DOI: https://doi.org/10.7554/eLife.43257.012

*Figure 3 continued on next page*

*Figure 3 continued*

**Figure supplement 2.** Stabilizing beta-catenin in ECs switches posterior pituitary ECs more than anterior pituitary ECs to a BBB-like state.
DOI: https://doi.org/10.7554/eLife.43257.013

**Figure supplement 3.** *Pdgfb-CreER* efficiently recombines a *LoxP-stop-LoxP* reporter in both anterior and posterior pituitary ECs.
DOI: https://doi.org/10.7554/eLife.43257.014

posterior pituitary. *Figure 3E and F* show a similar analysis for MDR1 and OCLN in the posterior pituitary from mice with mosaic excision of *Ctnnb1* exon 3 in ~80% of ECs. The cellular distributions of MDR1 and OCLN closely match the distribution of CLDN5, which marks the ECs that excised *Ctnnb1* exon 3, and they are anti-correlated with the distribution of PLVAP. Importantly, the subcellular distributions of CLDN5 and OCLN are also closely matched, and are consistent with their accumulation at junctional complexes (*Figure 3—figure supplement 1*). Curiously, the ratio of CLDN5 and OCLN intensities varies among the posterior pituitary ECs that express these proteins (*Figure 3—figure supplement 1D*).

Consistent with the accumulation of LEF1 in CVO ECs in response to beta-catenin stabilization, an EC-specific H2B-GFP reporter of beta-catenin signaling [*R26-8xTCF/LEF-loxP-stop-loxP(LSL)-H2B-GFP*; *Cho et al., 2017*] showed little or no H2B-GFP accumulation in WT ECs in the area postrema and strong nuclear accumulation in ECs in the area postrema following beta-catenin stabilization (arrows in the third pair of panels in *Figure 2—figure supplement 3*). As expected, the H2B-GFP reporter accumulates in the nuclei of BBB+ ECs in the surrounding CNS tissue in both WT and beta-catenin stabilized mice (arrowheads in the third pair of panels in *Figure 2—figure supplement 3*).

In the experiments in *Figure 2—figure supplement 3*, the EC specificity of the *R26-8xTCF/LEF-LSL-H2B-GFP* reporter results from the EC-specific excision of the *LSL* transcription stop cassette by *Pdgfb-CreER*. In *Figure 2—figure supplement 4*, the sensitivity and specificity of the *R26-8xTCF/LEF-LSL-H2B-GFP* reporter within CNS ECs is further validated by (1) the strong reduction in H2B-GFP accumulation in $Ndp^{KO}$ retinas [compare *Figure 2—figure supplement 4*, panel (A) vs. panel (D)] and in EC-specific *Fz4* KO retinas [compare *Figure 2—figure supplement 4*, panel (B) vs. panel (E)], and (2) the elevation in H2B-GFP levels in retinal ECs in a mouse with EC-specific beta-catenin stabilization compared to control retinas [compare *Figure 2—figure supplement 4*, panel (C) vs. panel (F)].

## Distinct responses of anterior and posterior pituitary ECs to beta-catenin stabilization

In WT mice, anterior pituitary ECs, like posterior pituitary ECs, express PLVAP and repress CLDN5, GLUT1, LEF1, MDR1, OCLN, and the CNS EC-specific docosahexaenoic acid transporter MFSD2A (*Ben-Zvi et al., 2014*; *Nguyen et al., 2014*; left panels of *Figure 3* and *Figure 3—figure supplement 2*). In response to beta-catenin stabilization, posterior pituitary ECs repressed PLVAP and activated CLDN5, GLUT1, MFSDA, and LEF1. By contrast, anterior pituitary ECs showed a more modest reduction in the level of PLVAP and a more modest increase in CLDN5 expression, and the levels of GLUT1, MFSD2A, and LEF1 remained below the limits of detection (right panels of *Figure 3* and *Figure 3—figure supplement 2*). This differential response does not reflect differential activity of the *Pdgfb-CreER* driver in anterior vs. posterior pituitary ECs, as *Pdgfb-CreER* drives highly efficient recombination of a Cre-reporter (*R26-LSL-tdTomato-2A-H2B-GFP*) in ECs in both the anterior and posterior divisions of the pituitary with or without beta-catenin stabilization (*Figure 3—figure supplement 3*).

The contrasting responses of anterior vs. posterior pituitary ECs to beta-catenin stabilization could reflect the different embryonic origins of these two neuroendocrine tissues. The posterior pituitary is derived from the neural tube and is embryologically part of the CNS, whereas the anterior pituitary is derived from Rathke's pouch and is therefore a derivative of non-neural epithelium. As described below in connection with Figures 6–9, we address this issue with an analysis of the full transcriptomes of anterior and posterior pituitary ECs, with and without beta-catenin stabilization.

## Response of specialized ocular vasculatures to beta-catenin stabilization

As noted in the Introduction, the ECs of the choriocapillaris and the ciliary body are highly permeable. Immunolocalization of CLDN5, PLVAP, and GLUT1 in sections of whole eyes shows that WT ciliary body ECs are CLDN5-/PLVAP+ (*Figure 4A and A'*). The epithelial cells that surround the ciliary body capillaries are GLUT1+. In contrast, ECs in the retinal vasculature are CLDN5+/PLVAP-, with weak expression of GLUT1 (*Figure 4C and C'*), consistent with their role in the BRB. The ECs that comprise the choriocapillaris are CLDN5-/PLVAP+, and the immediately adjacent RPE expresses GLUT1 at high levels (*Figure 4C,C' and C''*). These observations are consistent with the current picture of choriocapillaris and RPE cooperation, in which small molecules, such as glucose, can freely permeate the highly fenestrated choriocapillaris but must be actively transported across the RPE to gain access to the retina (*Alm, 1992*).

Beta-catenin stabilization in ECs leads to a phenotypic conversion of ciliary body and choriocapillaris ECs that is analogous to the conversion described above for CVO and choroid plexus ECs. Ciliary body and choriocapillaris ECs convert in an all-or-none fashion from CLDN5-/PLVAP+ to CLDN5+/PLVAP- (*Figure 4A'* vs. B', and C' vs. D'; the choriocapillaris shown in D' and D' is mosaic for *Pdgfb-CreER*-mediated recombination). There is no change in these markers in the retinal vasculature (*Figure 4D and D'*). Low magnification images of the retina and choroid in cross section (*Figure 4E and F*) and of the choroid and RPE as a flat mount without the neural retina (*Figure 4G and H*) show the conversion of choriocapillaris ECs to a CLDN5+/PLVAP- state upon beta-catenin stabilization. The conversion of ciliary body ECs is less efficient.

Taken together, these observations imply that the EC gene expression program governing capillary permeability within the ciliary body and choriocapillaris can be regulated by beta-catenin signaling in a manner analogous to the regulation described above for CVOs and the choroid plexus. In all of these vasculatures, mature ECs retain substantial plasticity with respect to this gene expression program, so that an acute increase in beta-catenin signaling can suppress one set of markers (e.g. PLVAP) and induce another set (e.g. CLDN5). There are also differences among capillary beds in the responses of individual BBB/BRB markers to beta-catenin stabilization. For example, beta-catenin stabilization induces GLUT1 in posterior pituitary ECs but not in choriocapillaris ECs.

## Ultrastructural and permeability effects of beta-catenin stabilization in brain and ocular ECs

The most obvious ultrastructural distinction between permeable and non-permeable CNS ECs is the presence of fenestrae in permeable ECs and their absence on non-permeable ECs. In transmission electron microscope (TEM) images, each CVO fenestra has a single diaphragm that spans the transcellular opening and that appears as a thin line of up to ~60 nm in length (*Stan et al., 2012*). In cross-sections, fenestrae typically appear in groups of ~5 to ~20, and in freeze-fracture (i.e. en face) images they typically appear in groups of ~50 to ~100 with relatively uniform center-to-center spacing (*Maul, 1971*). Fenestrae are associated with regions where the EC component of the capillary wall is extremely thin, with EC luminal and basal plasma membranes separated by ~100 nm. In contrast, the EC component of the capillary wall in non-fenestrated CNS capillaries is typically several hundred nm in thickness. PLVAP has been identified as the major (and perhaps the only) structural protein that comprises the fenestral diaphragm (*Stan et al., 2004*), and therefore its presence in permeable ECs and its absence from non-permeable CNS ECs correlates with the presence and absence of fenestrae, respectively.

To assess the effect of beta-catenin signaling at the ultrastructural level, the choroid plexus, choriocapillaris, ciliary body, and posterior pituitary were analyzed from control mice (n = 3–5) and from EC-specific beta-catenin stabilized mice (n = 3–4). *Figure 5A–F* compares representative electron micrographs from control capillaries (left) and EC-specific beta-catenin stabilized capillaries (right), and *Figure 5G* quantifies the number of fenestrae per unit length of capillary wall. In each of the four capillary beds, stabilization of beta-catenin led to a substantial loss of fenestrae. The non-zero density of fenestrae in the *Ctnnb1^{flex3/+}*;*Pdgfb-CreER* samples likely reflects (at least in part) a minority of ECs in which *Pdgfb-CreER* failed to recombine the *Ctnnb1^{flex3}* target.

To assess the effect of beta-catenin signaling on vascular permeability in the CVOs, Sulfo-N-hydroxysuccinimide (NHS)-biotin, a low molecular weight tracer that reacts covalently with free amines, was introduced into the intravascular space by intraperitoneal (IP) injection ten minutes prior

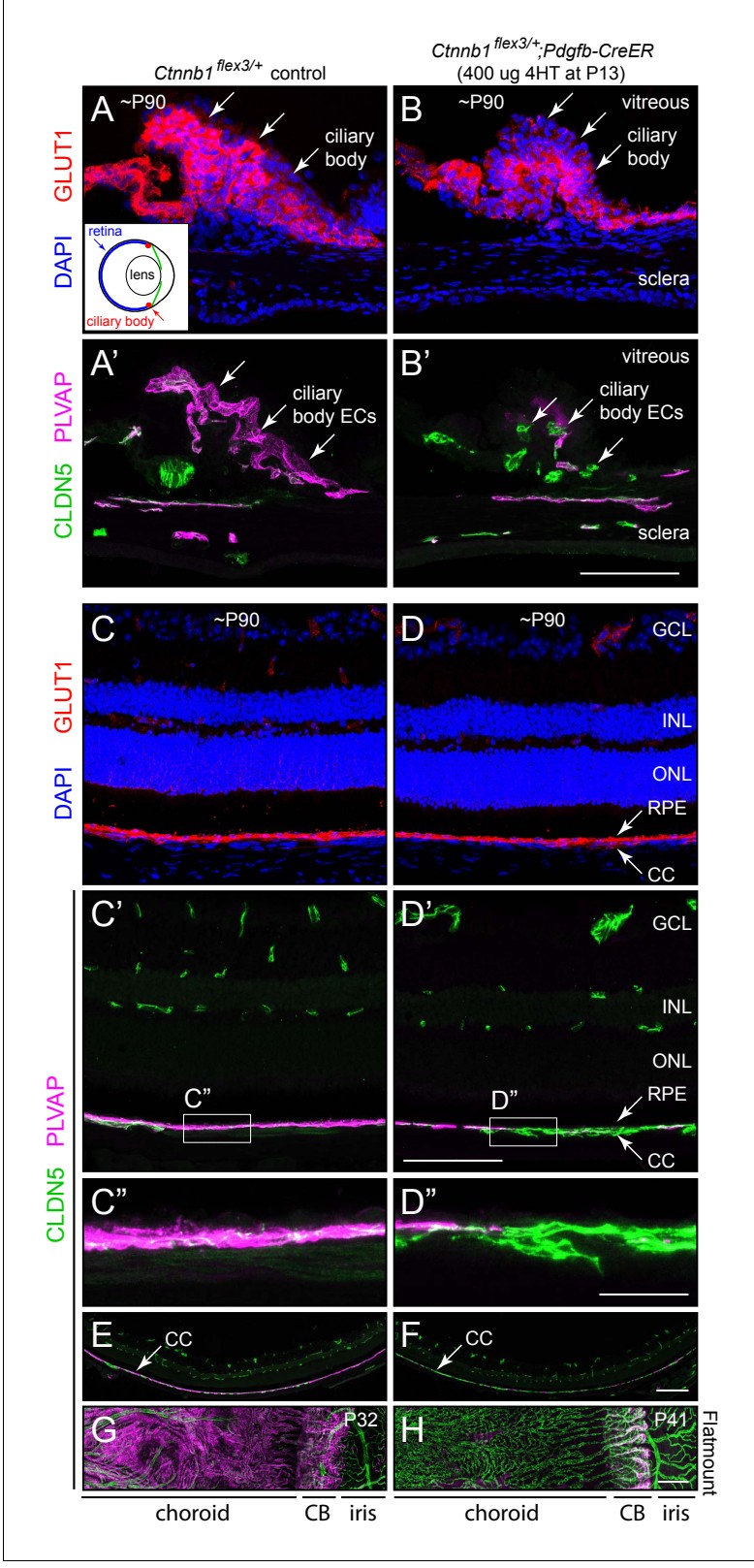

**Figure 4.** Stabilizing beta-catenin in ECs switches the choriocapillaris and ciliary body vasculatures to a BBB-like state. (A–B') Cross-sections of the ciliary body from a control mouse (**A, A'**) and from a mouse with mosaic EC-specific deletion of *Ctnnb1* exon 3 (**B, B'**), immunostained as indicated. GLUT1 is expressed in epithelial cells surrounding the ciliary body capillaries. ECs in the control ciliary body are CLDN5-/PLVAP +and ECs with deletion

*Figure 4 continued on next page*

*Figure 4 continued*
of *Ctnnb1* exon 3 are CLDN5+/PLVAP- (arrows in A' and B'). Inset in (A), schematic of the mouse eye showing the locations of the ciliary body, lens, and retina. Scale bar, 100 μm. (C–D") Cross-sections of retina from a control mouse (C, C', C") and from a mouse with mosaic EC-specific deletion of *Ctnnb1* exon 3 (D, D, D"), immunostained as indicated. With EC-specific deletion of *Ctnnb1* exon 3, choriocapillaris ECs switch from CLDN5-/PLVAP +to CLDN5+/PLVAP-. The adjacent RPE is GLUT1+. Intra-retinal ECs remain CLDN5+/PLVAP-. (C") and (D") show higher magnification views of the boxed regions from panels (C') and (D'). GCL, ganglion cell layer; INL, inner nuclear layer; ONL, outer nuclear layer; RPE, retinal pigment epithelium; CC, choriocapillaris. Scale bar for C-D', 100 um. Scale bar for C' and D', 20 μm. (E,F) Lower magnification images encompassing the regions shown in (C') and (D'). Scale bar, 200 um. (G,H) Flat mount of the retinal pigment epithelium, choroid, and sclera after the retina was removed, viewed from the anterior of the eye. With EC-specific deletion of *Ctnnb1* exon 3, capillary ECs in the choriocapillaris switch from CLDN5-/PLVAP +to CLDN5+/PLVAP-. The ciliary body shows less complete conversion. Scale bar, 200 μm.
DOI: https://doi.org/10.7554/eLife.43257.015

to intracardiac perfusion in age-matched WT and *Ctnnbl1*$^{flex3/+}$;*Pdgfb-CreER* mice. Covalently immobilized Sulfo-NHS-biotin accumulation was subsequently visualized with fluorescent Streptavidin. Beta-catenin stabilization in *Ctnnbl1*$^{flex3/+}$;*Pdgfb-CreER* mice resulted in the expected conversion of most of the CVO and CP ECs from a GLUT1-/PLVAP+ state to a GLUT+/PLVAP- state, and this conversion was accompanied by a dramatic reduction in the amount of perivascular Sulfo-NHS-biotin in the area postrema and SFO, with a more modest reduction in the choroid plexus (*Figure 5H and I*).

In summary, these analyses show that activation of beta-catenin signaling in ECs leads not only to alterations in the expression of molecular markers, but also to the morphologic conversion of ECs from a fenestrated to a non-fenestrated phenotype together with a reduction in vascular permeability.

## EC transcriptomes with and without beta-catenin stabilization

To obtain a genome-wide view of transcripts and chromatin accessibility associated with activation of beta-catenin signaling in high permeability CNS vascular beds, we applied RNA-seq and Assay for Transposase-Accessible Chromatin (ATAC)-seq (*Buenrostro et al., 2013*) to FACS-purified ECs from the cerebellum, the anterior pituitary, and the posterior pituitary of phenotypically WT *Tie2-GFP* (the *Tie2* gene is also known as *Tek*) adult mice and EC-specific beta-catenin stabilized *Ctnnbl1*$^{flex3/+}$; *Pdgfb-CreER*;*Tie2-GFP* adult mice. The cerebellum serves here as a representative example of BBB + CNS, the posterior pituitary as a representative CVO, and the anterior pituitary as a non-CNS tissue with highly permeable vasculature. We chose the pituitary over other CVOs for three reasons: (1) unlike other CVOs that are embedded within the brain, the pituitary protrudes from the brain and can be dissected free from other tissues; (2) the anterior and posterior lobes of the pituitary can be cleanly separated from one another; and (3) the vascular beds of the anterior and posterior pituitary exhibit different responses to beta-catenin stabilization, as detailed above.

In both anterior and posterior pituitary ECs, genome browser images show that beta-catenin stabilization results in: (1) increased expression of genes that are normally expressed in cerebellum ECs but not in pituitary ECs (e.g. *Slc35f2, Axin2, Mfsd2a,* and *Slc2a1*), with greater induction of these genes in posterior pituitary ECs compared to anterior pituitary ECs, and (2) increased chromatin accessibility in genomic regions that are normally accessible in cerebellum ECs but not in pituitary ECs (compare tracks labeled WT vs. Bcat in *Figure 6A* and *Figure 6—figure supplement 1*; chromatin regions of interest are highlighted in pale orange). *Figure 6—figure supplement 1* shows the high reproducibility of independent replicate samples. As seen in *Figure 6A*, *Figure 6—figure supplement 1*, and in the figures that follow, a consistent feature of BBB transcript levels and ATAC peak areas in beta-catenin stabilized pituitary ECs is that, even when induced, the majority are substantially lower than their counterparts in cerebellar ECs.

WT cerebellum ECs, anterior pituitary ECs, and posterior pituitary ECs exhibited similar abundances of generic EC transcripts; for this group of transcripts, the principal effect of beta-catenin stabilization was a modest reduction in abundance in posterior pituitary ECs [*Figure 6B*, top row, 'General vascular EC transcripts'; RNA read counts are shown as transcripts per million (TPM)]. 224 transcripts were enriched in WT cerebellum ECs compared to WT anterior pituitary and WT posterior pituitary

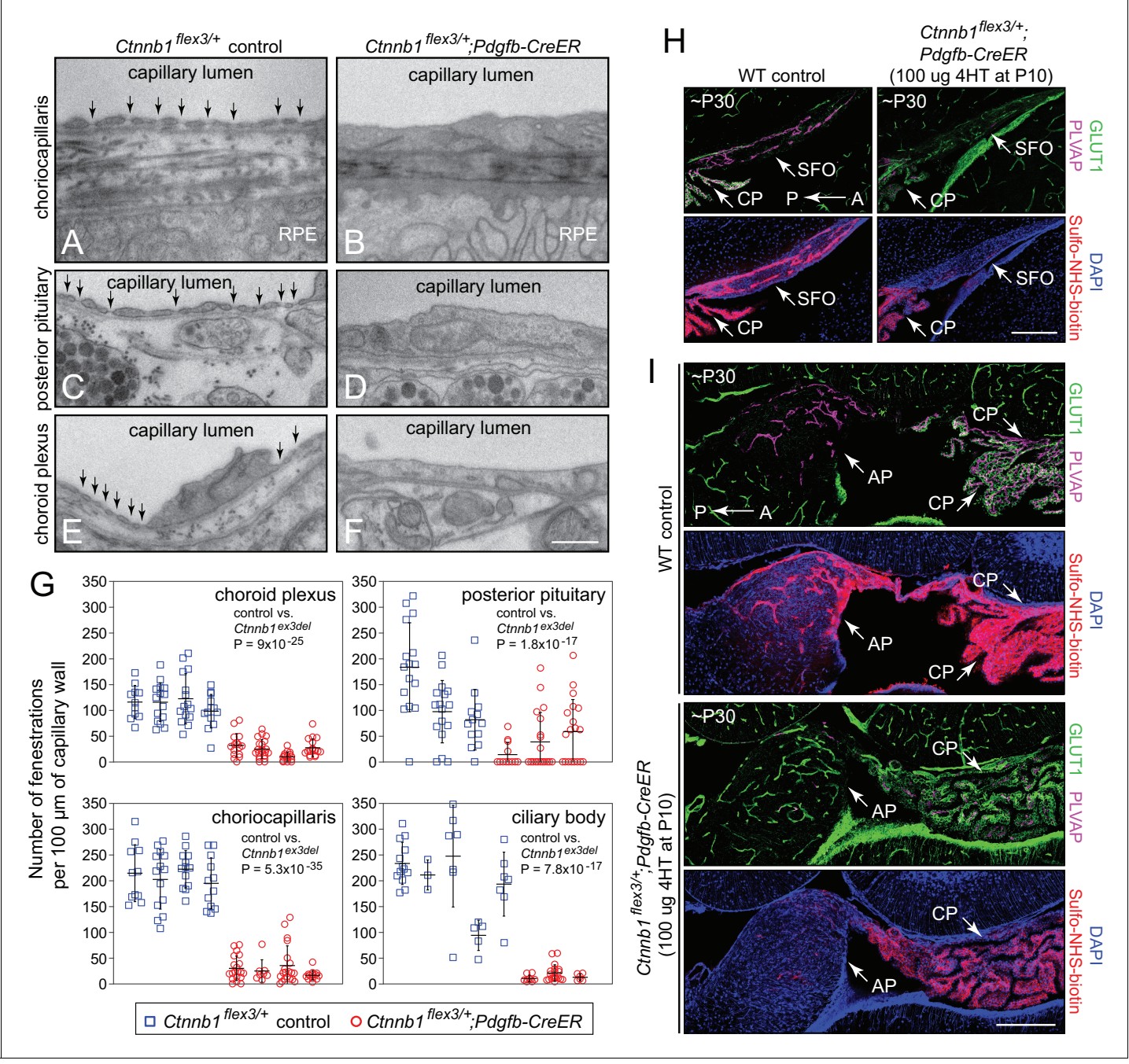

**Figure 5.** Stabilizing beta-catenin signaling suppresses EC fenestrae and reduces vascular permeability. (**A–F**) Transmission electron micrographs from control mice (left) and from age-matched mice with EC-specific deletion of *Ctnnb1* exon 3 (right) showing representative sections of choriocapillaris (**A, B**), posterior pituitary (**C,D**), and choroid plexus (**E,F**). RPE, retinal pigment epithelium. Vertical arrows mark fenestrae. Neurohormone secretory granules are seen in the posterior pituitary parenchyma. Scale bar, 500 nm. (**G**) Quantification of the density of capillary EC fenestrae in age-matched and 4HT-treated control (*Ctnnb1^{flex3/+}*) vs. *Ctnnb1* exon 3 stabilized (*Ctnnb1^{flex3/+};Pdgfb-CreER*) mice for the choriocapillaris, posterior pituitary, choroid plexus, and ciliary body. Each data-point represents all of the vascular wall length within a single 10 µm x 10 µm TEM image. Bars show mean ± S.D. Each cluster of data points represents a different mouse. For each location and genotype, 3–5 mice were analyzed. For each of the four anatomic locations, the p-values are calculated for all of the *Ctnnb1^{flex3/+}* vs. all of the *Ctnnb1^{flex3/+};Pdgfb-CreER* data-points. (**H,I**) Sagittal brain sections show vascular markers GLUT1 and PLVAP and perivascular accumulation of Sulfo-NHS-biotin (following IP injection) in the SFO and adjacent choroid plexus (**H**) and in the area postrema and adjacent choroid plexus (**I**) from ~P30 WT vs. *Ctnnb1^{flex3/+};Pdgfb-CreER* mice. AP, area postrema; CP, choroid plexus. A, anterior; P, posterior. Scale bars in (**H**) and (**I**), 200 µm.
DOI: https://doi.org/10.7554/eLife.43257.016

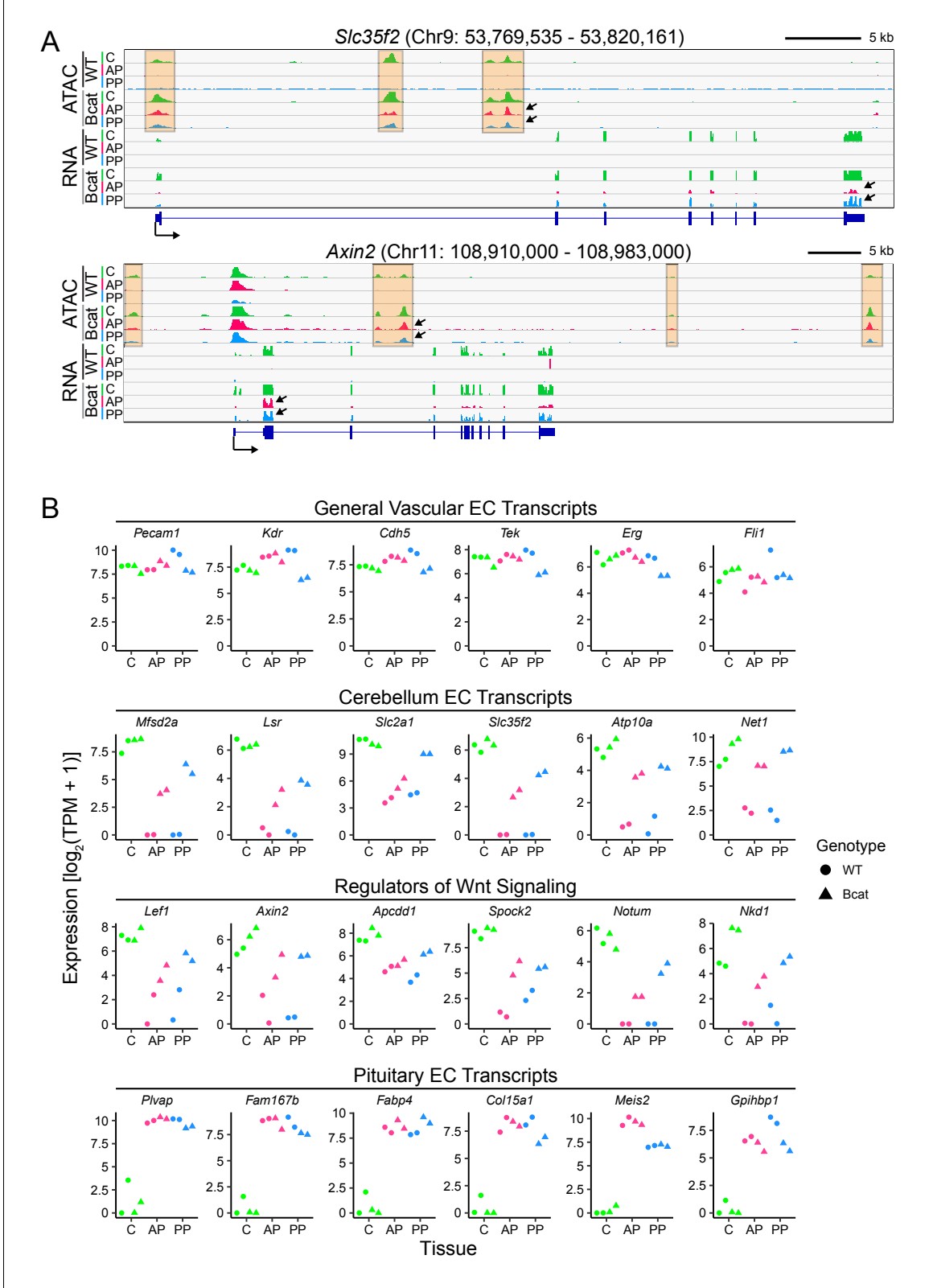

**Figure 6.** In pituitary ECs, stabilization of beta-catenin promotes a BBB-like state of the transcriptome and of chromatin accessibility. (**A**) Genome browser images showing accessible chromatin (ATAC; top) and transcript abundances (RNA; bottom) for two BBB genes: *Slc35f2*, which codes for a putative nucleoside transporter, and *Axin2*, a Wnt target gene and a negative regulator of Wnt signaling. Histograms show aligned read counts. C, cerebellum; AP, anterior pituitary; PP, posterior pituitary; WT, wild-type; Bcat, beta-catenin stabilized (i.e. *Ctnnb1* exon 3 deleted). Each track represents
*Figure 6 continued on next page*

*Figure 6 continued*

the average of two independent replicates. The orange regions highlight accessible chromatin that changes with beta-catenin stabilization. For each gene, all six of the ATAC-seq histograms are at the same vertical scale [*Slc35f2* ATAC: 0–2000 reads; *Axin2* ATAC: 0–3000 reads] and all six of the RNA-seq histograms are at the same vertical scale [*Slc35f2* RNA: 0–5251 reads; *Axin2* RNA: 0–3379 reads]. For both genes, some of the ATAC-seq and RNA-seq signals extend beyond the vertical range in the images. Bottom, intron-exon structure, with the arrow indicating the promoter and the direction of transcription. Arrows within the browser images point to examples of ATAC-seq or RNA-seq signals that are induced by beta-catenin stabilization. (B) Expression levels (log$_2$ transformed TPM + 1) based on RNA-seq for the indicated categories of EC transcripts. Data for the two independent replicates are shown.

DOI: https://doi.org/10.7554/eLife.43257.017

The following figure supplements are available for figure 6:

**Figure supplement 1.** Genome browser images for two genomic loci that are responsive to beta-catenin stabilization in anterior and posterior pituitary ECs.

DOI: https://doi.org/10.7554/eLife.43257.018

**Figure supplement 2.** Heatmap showing the abundances of transcripts coding for tight junction proteins in WT vs.beta-catenin stabilized ECs in pituitary and cerebellum.

DOI: https://doi.org/10.7554/eLife.43257.019

**Figure supplement 3.** Heatmap showing the abundances of transcripts coding for TFs in WT vs. beta-catenin stabilized ECs in pituitary and cerebellum.

DOI: https://doi.org/10.7554/eLife.43257.020

ECs (*Supplementary file 1*; here and elsewhere, the cut-off for 'enrichment' is 2-fold, unless stated otherwise). We will refer to these as 'cerebellar BBB transcripts'. Among these 224 cerebellar BBB transcripts, 31% and 45% increased in abundance >2 fold upon beta-catenin stabilization in anterior and posterior pituitary ECs, respectively, compared to the corresponding WT pituitary EC controls (*Supplementary file 1*). In the second and third rows of *Figure 6B*, TPM are plotted for twelve of these transcripts; six were previously shown to be enriched both in ECs compared to non-ECs and in CNS compared to non-CNS ECs [(*Sabbagh et al., 2018*); second row, 'Cerebellum EC transcripts'] and six are known Wnt-responsive transcripts that are also enriched in CNS compared to non-CNS ECs [(*Sabbagh et al., 2018*); third row, 'Regulators of Wnt signaling']. For genes in these two categories, the level of expression with beta-catenin stabilization is, on average, several-fold greater in posterior pituitary ECs than in anterior pituitary ECs.

93 transcripts were enriched in WT anterior pituitary ECs compared to WT posterior pituitary ECs and WT cerebellum ECs; 110 transcripts were enriched in WT posterior pituitary ECs compared to WT anterior pituitary ECs and WT cerebellum ECs; and 158 transcripts were enriched in common between WT anterior pituitary ECs and WT posterior pituitary ECs compared to WT cerebellum ECs (*Supplementary file 1*). The RNA TPMs for six of these transcripts are plotted in the last row of *Figure 6B* ('Pituitary EC transcripts'). In contrast to the 31–45% of cerebellar BBB transcripts that increased in abundance in pituitary ECs upon beta-catenin stabilization, as described in the preceding paragraph, only 9% and 20% of pituitary EC transcripts decreased in abundance in anterior and posterior pituitary ECs, respectively, upon beta-catenin stabilization. This asymmetry could be biologically relevant or it could be, at least partly, explained by incomplete Cre-mediated recombination, since gene expression that is induced by beta-catenin from a low baseline in a mosaic population of ECs would likely show >2 fold induction (and, therefore, be counted as 'enriched'), whereas a decrease in gene expression would be partially obscured by transcripts derived from unrecombined ECs. The transcriptome analysis also suggests that post-translational mechanisms add additional specificity to the BBB vs. non-BBB decision, as seen in comparing the modest reduction in Plvap transcript levels (*Figure 6B*) to the dramatic reduction in PLVAP protein (*Figure 3*) in beta-catenin stabilized posterior pituitary ECs.

Among transcripts coding for tight junction proteins in posterior pituitary ECs, those coding for CLDN5 and OCLN increase in abundance upon beta-catenin stabilization, as do *Lsr* transcripts (coding for the tricellular tight junction protein LSR), but *Tjp1* transcripts (coding for ZO1) show a more modest response, and seven transcripts coding for tight junction proteins that are not specifically associated with BBB ECs show no significant changes (*Figure 6—figure supplement 2*). Among transcripts coding for transcription factors (TFs) that are specifically enriched in cerebellar ECs, only 7/14 were induced by beta-catenin stabilization (red labels in *Figure 6—figure supplement 3*). For

*Foxq1* transcripts induction is greater in the posterior pituitary than in the anterior pituitary, whereas for *Etv6* transcripts induction is greater in the anterior pituitary than in the posterior pituitary. Transcripts coding for pan-EC TFs show a modest decline in abundance in both anterior and posterior pituitary upon beta-catenin stabilization.

## Genome-wide analyses of EC transcriptomes

To globally visualize differences in gene expression, we generated scatter plots of cross-sample normalized RNA-seq read counts for the set of 2479 transcripts in the six EC samples that showed >2 fold enrichment compared to their surrounding non-EC (GFP-negative) parenchymal cells, which were also FACS-purified and characterized by RNA-seq; we refer to this set as 'EC-enriched'. *Figure 7A* highlights, by color, the subsets of these transcripts that were enriched >2 fold in WT cerebellum ECs vs. WT anterior pituitary ECs (left), WT cerebellum ECs vs. WT posterior pituitary ECs (center), and WT anterior pituitary ECs vs. WT posterior pituitary ECs (right). [To assist the reader in following the trends of abundance changes for BBB-associated transcripts, seven of these transcripts are highlighted and labeled in each of the scatterplots in *Figure 7*.] Consistent with immunostaining (*Figure 3* and *Figure 3—figure supplements 1* and *2*) and genome browser images (*Figure 6A* and *Figure 6—figure supplement 1*), beta-catenin stabilization induces the cerebellar BBB gene expression program more strongly in posterior pituitary ECs compared to anterior pituitary ECs, and this is reflected in the tighter clustering of the colored data points along the 45-degree line in the plots of WT cerebellum ECs vs. beta-catenin stabilized posterior pituitary ECs compared with the plots of WT cerebellum ECs vs. beta-catenin stabilized anterior pituitary ECs (*Figure 7B*).

A comparison of cerebellar ECs with vs. without beta-catenin stabilization is shown in *Figure 7B* (right-most plots). With beta-catenin stabilization, cerebellum ECs, which normally exist in a state of constitutive beta-catenin activation, show little change in the abundances of cerebellar BBB transcripts – that is, these points reside close to the 45-degree line - but there is a clear reduction in the abundance of many anterior and posterior pituitary enriched EC transcripts, which reside above the 45-degree line (*Figure 7B*, right-most plots; vertical black arrows). Similarly, plots of transcript abundances for posterior pituitary ECs with vs. without beta-catenin stabilization and anterior pituitary ECs with vs. without beta-catenin stabilization show that pituitary-specific EC transcripts are minimally altered but cerebellar BBB transcripts are increased in abundance with beta-catenin stabilization (*Figure 7C*; vertical black arrows).

Importantly, in *Figure 7B* (four left plots) the shifts in the colored data points toward the 45 degree line do not simply arise from regression to the mean of outlying data points, as might be observed with independent samples containing noisy data. This is demonstrated in *Figure 7—figure supplement 1*, where the highlighted (i.e.,>2 fold enriched) data points are defined using either the first set of replicates (panels in A) or the second set of replicates (panels in B) and are then visualized with scatter plots derived from the first set of replicates (top row) or the second set of replicates (second row). It is apparent that the behavior of the two pairs of highlighted data points is nearly identical (and is nearly identical to the averaged data shown in *Figure 7*) regardless of whether the colored data points derive from the same or independent experiments. This near identity in behavior in cross-replicate comparisons is consistent with the high correlation between replicates (lower two rows in *Figure 7—figure supplement 1*).

A principal component analysis (PCA) of the expression levels of 2466 WT EC-enriched genes (i.e. enriched >2 fold in WT ECs compared to their surrounding non-EC parenchymal cells) shows that (1) the transcriptomes of the EC replicates cluster tightly by tissue-of-origin and genotype, and (2) beta-catenin stabilization shifts each transcriptome in the same direction (downward) along PC1, which also shifts the pituitary EC transcriptomes toward the cerebellar EC transcriptome (*Figure 7D*). If we consider only the 427 WT EC-enriched transcripts that were enriched in a single tissue (224 cerebellum EC + 93 anterior pituitary EC + 110 posterior pituitary EC = 427 total), a heat-map of Pearson correlations shows that: (1) compared to WT posterior pituitary ECs, beta-catenin stabilized posterior pituitary ECs have a substantially higher correlation with cerebellum ECs, and (2) compared to WT anterior pituitary ECs, beta-catenin stabilized anterior pituitary ECs have a modestly higher correlation with cerebellum ECs (*Figure 7E*). In sum, the analyses in *Figures 6* and *7* show that the main consequence of beta-catenin stabilization in pituitary ECs is to increase the abundance of BBB and other beta-catenin-responsive transcripts, and the main consequence in cerebellar ECs is to reduce the abundance of non-BBB EC transcripts.

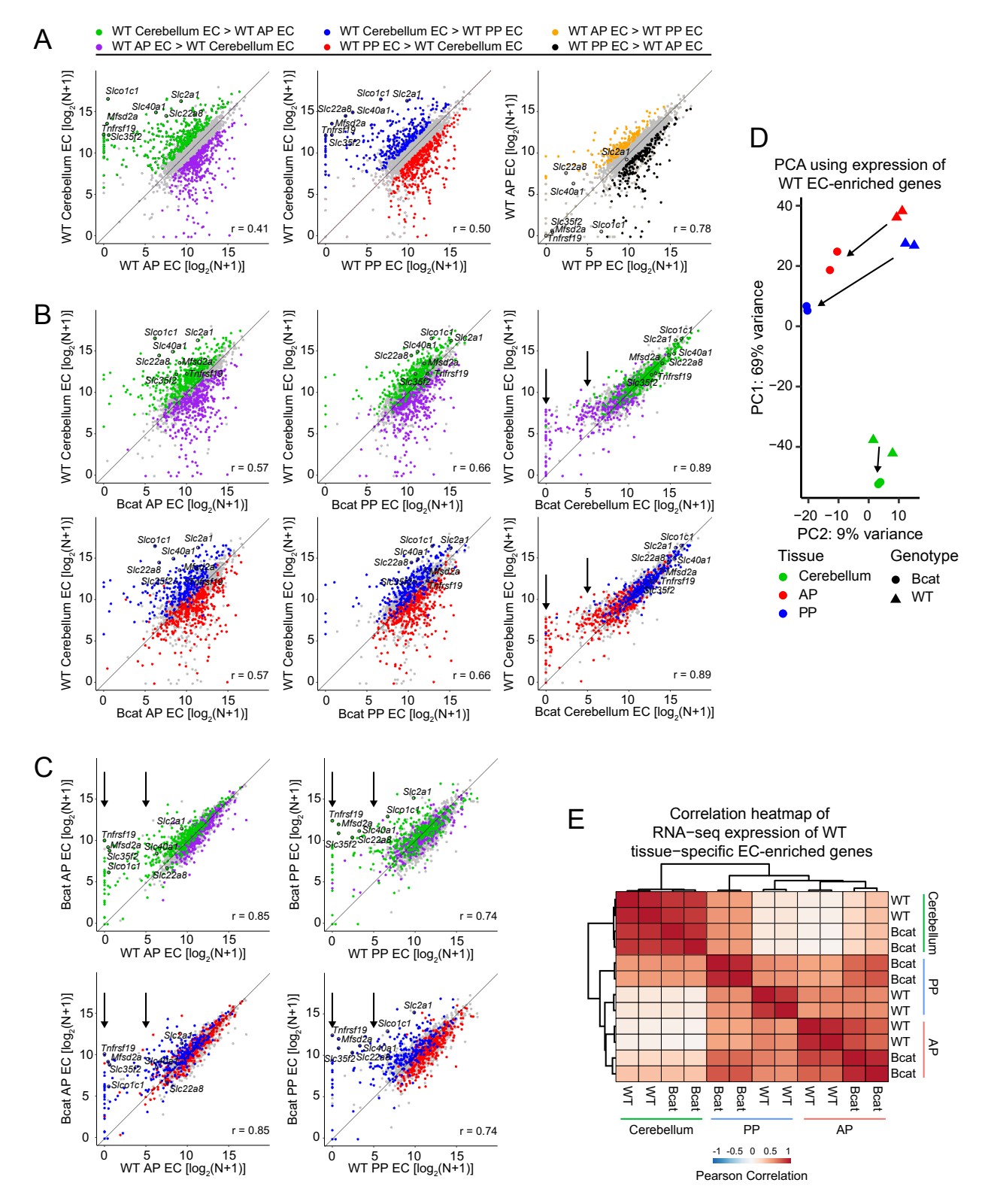

**Figure 7.** Genome-wide visualization of transcriptome differences in ECs from different tissues for WT vs.beta-catenin stabilized samples. (**A**) Scatter plots comparing cross-sample normalized RNA-seq read counts of EC-enriched protein-coding genes from cerebellum ECs vs. anterior pituitary (AP) ECs, cerebellum ECs vs. posterior pituitary (PP) ECs, and anterior pituitary ECs vs. posterior pituitary ECs. Values depicted are the log₂ transformation of cross-sample normalized RNA-seq read counts + 1. Colored symbols indicate transcripts with false discovery rate (FDR) < 0.05 and enrichment >2

*Figure 7 continued on next page*

*Figure 7 continued*

fold for the indicated comparisons. The same set of seven BBB-associated transcripts are labeled in each scatterplot and their associated datapoints are enclosed by black circles. All data in this figure are averages of two independent samples. (B) Scatter plots comparing cross-sample normalized RNA-seq read counts of EC-enriched protein-coding genes from WT cerebellum ECs vs. beta-catenin stabilized anterior pituitary ECs, posterior pituitary ECs, and cerebellum ECs. Colored symbols are as defined in (A). (C) Scatter plots comparing cross-sample normalized RNA-seq read counts of EC-enriched protein-coding genes between genotypes for anterior pituitary ECs and posterior pituitary ECs. Vertical arrows in (B) and (C) indicate multiple data points that change upon beta-catenin stabilization. (D) Principal component analysis of all WT EC-enriched transcripts from cerebellum ECs, anterior pituitary ECs, and posterior pituitary ECs. The two symbols for each sample represent biological replicates. Most of the variance is explained by PC1 which separates the samples by tissue of origin. Arrows indicate changes upon beta-catenin stabilization. (E) Heatmap indicating pairwise Pearson correlations for RNA-seq TPM among WT and beta-catenin-stabilized ECs for tissue-specific EC protein-coding genes. Data are shown for the individual replicates. AP, anterior pituitary; PP, posterior pituitary.

DOI: https://doi.org/10.7554/eLife.43257.021

The following figure supplements are available for figure 7:

**Figure supplement 1.** Genome-wide visualization of transcriptome differences in WT vs. beta-catenin stabilized ECs, comparing across independent replicates.
DOI: https://doi.org/10.7554/eLife.43257.022

**Figure supplement 2.** Genome-wide visualization of transcriptome differences in choroid plexus ECs vs. cerebral cortex ECs in response to beta-catenin stabilization, as determined by RiboTag enrichment.
DOI: https://doi.org/10.7554/eLife.43257.023

To complement the pituitary EC vs. cerebellum EC comparisons, we utilized RiboTag technology to compare EC transcripts associated with polyribosomes in ECs in the cerebral cortex and choroid plexus of WT and beta-catenin stabilized adult mice. [Ribotag mice have been genetically engineered to express an epitope-tagged ribosomal protein in a Cre-recombinase-dependent manner, which permits cell-type-specific immuno-affinity capture of ribosome-associated mRNAs (*Sanz et al., 2009*). For these experiments we compared $Rpl22^{ribotag/+}$; $Ctnnb1^{flex3/+}$;*Pdgfb-CreER* vs. $Rpl22^{ribotag/+}$;*Pdgfb-CreER* mice several weeks after an IP injection of 4HT.] Based on RiboTag RNA-seq, 184 transcripts were enriched in WT cortex ECs compared to WT choroid plexus ECs ('cortical BBB transcripts') and 311 transcripts showed the reverse enrichment (*Supplementary file 2*). Among the 184 cortical BBB transcripts, 28% were increased >2 fold upon beta-catenin stabilization in choroid plexus ECs. Scatter plots analogous to those described above for *Figure 7* show that beta-catenin stabilization in choroid plexus ECs partially induces the BBB gene expression program, that is, choroid plexus ECs become more like cortex ECs (*Figure 7—figure supplement 2A*). A cross comparison between the cerebellum EC vs. pituitary EC and the cortex EC vs. choroid plexus EC experiments reveals a substantial sharing of up- and down-regulated genes in response to beta-catenin stabilization (*Figure 7—figure supplement 2B*).

In summary, beta-catenin stabilization in ECs leads to a partial conversion to a BBB-like gene expression program in anterior pituitary, posterior pituitary, and choroid plexus ECs.

## Accessible chromatin and transcription factor binding motifs in ECs, with and without beta-catenin stabilization

Based on pair-wise comparisons among the three WT samples, 7813, 5714, and 4663 ATAC-seq peaks were more accessible (i.e., showed >2 fold more read counts) in WT cerebellum ECs, WT anterior pituitary ECs, and WT posterior pituitary ECs, respectively, in at least one pair-wise comparison (*Supplementary file 3*). These peaks are highlighted in color in the scatter plots in *Figure 8A*. A subset of these peaks are uniquely associated with a given EC-subtype (i.e. enriched >2 fold in both pairwise comparisons): 2680, 1493, and 84 in WT cerebellum ECs, WT anterior pituitary ECs, and WT posterior pituitary ECs, respectively (*Supplementary file 3*). Akin to the RNA-seq analyses described above, in both sets of pituitary ECs, beta-catenin stabilization induced an accessible chromatin state that was more like the state of cerebellar ECs, that is, the BBB-competent EC state is characterized by enhanced accessibility of cerebellar EC ATAC-seq peaks and reduced accessibility of pituitary EC ATAC-seq peaks (*Figure 8B*; top two rows). Cerebellum ECs, which – as noted above – normally exist in a state of constitutive beta-catenin activation, showed minimal changes in chromatin accessibility with beta-catenin stabilization (*Figure 8B*; third row). Also similar to the RNA-seq analyses, PCA of read densities at all called ATAC-seq peaks shows that (1) the EC replicates cluster

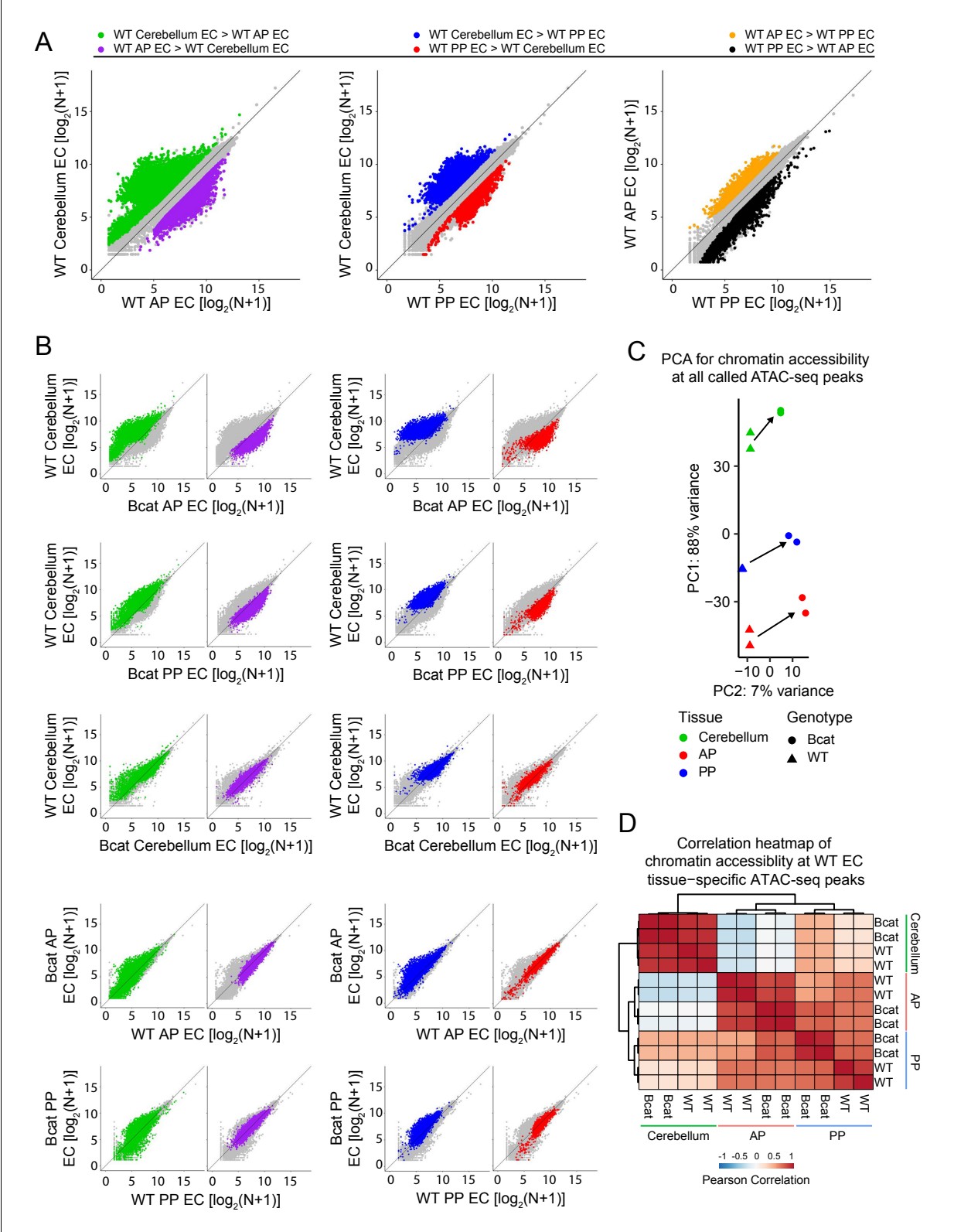

**Figure 8.** Genome-wide visualization of accessible chromatin differences by EC tissue type in WT vs. beta-catenin stabilized samples. (**A**) Scatter plots comparing cross-sample normalized ATAC-seq read counts in ATAC-seq peaks (i.e. peaks above threshold; see Materials and methods) from cerebellum ECs vs. anterior pituitary (AP) ECs, cerebellum ECs vs. posterior pituitary (PP) ECs, and anterior pituitary ECs vs. posterior pituitary ECs. Values depicted are the $\log_2$ transformation of cross-sample normalized ATAC-seq read counts + 1. Colored symbols indicate peaks with FDR < 0.05
*Figure 8 continued on next page*

*Figure 8 continued*

and differential accessibility >2 fold for the indicated comparisons. (B) Scatter plots of cross-sample normalized ATAC-seq read counts in ATAC-seq peaks for the indicated pairwise comparisons. Colored symbols are as defined in (A). (C) Principal component analysis of chromatin accessibility at all called ATAC-seq peaks from WT and beta-catenin stabilized cerebellum ECs, anterior pituitary ECs, and posterior pituitary ECs. The two symbols for each sample represent biological replicates. Most of the variance is explained by PC1 which separates the samples by tissue of origin. Arrows indicate changes upon beta-catenin stabilization. (D) Heatmap indicating pairwise Pearson correlations for chromatin accessibility at WT and beta-catenin-stabilized tissue-specific EC ATAC-seq peaks. Data are shown for the individual replicates. AP, anterior pituitary; PP, posterior pituitary.

DOI: https://doi.org/10.7554/eLife.43257.024

tightly by tissue-of-origin and genotype, and (2) beta-catenin stabilization shifts all data points in the same direction, which shifts the points representing pituitary ECs in the direction of the points representing cerebellar ECs (*Figure 8C*). Finally, if we consider only the 4257 ATAC-seq peaks that were enriched in WT ECs from a single tissue (2680 cerebellum + 1493 anterior pituitary + 84 posterior pituitary = 4257 total), a heat-map of Pearson correlations shows that (1) compared to WT posterior pituitary, beta-catenin stabilized posterior pituitary ECs have a higher correlation with cerebellum ECs, (2) compared to WT anterior pituitary, beta-catenin stabilized anterior pituitary ECs have a higher correlation with cerebellum ECs, and (3) the posterior pituitary EC vs. cerebellum EC correlations are higher than the anterior pituitary EC vs. cerebellum EC correlations (*Figure 8D*).

In beta-catenin stabilized ECs, the appearance of newly accessible chromatin in the neighborhood of newly activated genes suggests a causal relationship (*Figure 6A* and *Figure 6—figure supplement 1*). To explore this relationship, correlations between changes in gene expression and changes in accessible chromatin were quantified genome-wide by determining the percentage of accessible regions gained or lost upon beta-catenin stabilization within 100 kb of (1) EC tissue-specific genes, and (2) EC genes down- or up-regulated by beta-catenin stabilization (*Figure 9A*). A total of 1042, 2829, and 2060 chromatin regions gained accessibility – that is showed >2 fold more read counts – in beta-catenin stabilized cerebellar, anterior pituitary, and posterior pituitary ECs, respectively, whereas 1510, 1397, and 1100 regions of chromatin lost accessibility (*Supplementary file 3*). [As noted above in the context of the EC transcriptome analyses, the number of accessible chromatin regions gained or lost upon beta-catenin stabilization could be, at least partly, biased by incomplete Cre-mediated recombination since a region of chromatin accessibility that is gained relative to a low baseline would likely show a > 2 fold increase in read counts, whereas a loss in chromatin accessibility might be obscured by read counts from unrecombined ECs.] This analysis shows that accessible chromatin regions gained in beta-catenin stabilized ECs are significantly enriched within 100 kb of genes that are WT cerebellum EC-specific and that were induced by beta-catenin stabilization in anterior and posterior pituitary ECs. Moreover, the regions of accessible chromatin in anterior and posterior pituitary ECs that were gained upon beta-catenin stabilization exhibit a high degree of overlap with accessible chromatin regions that are unique to WT cerebellum ECs (*Figure 9B*; *Supplementary file 3*). In sum, beta-catenin stabilization in anterior and posterior pituitary ECs leads to chromatin remodeling in the neighborhood of cerebellar BBB genes, which makes the landscape of chromatin accessibility more like that of cerebellum ECs.

To link the accessible chromatin analyses to the actions of individual TFs, we used Hypergeometric Optimization of Motif EnRichment (HOMER), a suite of tools for discovering enriched motifs in genomic sequences (*Heinz et al., 2010*), to perform enrichment analyses of TF DNA binding motifs within accessible chromatin regions that (1) are specific to each EC subtype, and (2) were gained within each EC subtype upon beta-catenin stabilization (*Figure 9C*). For TFs that form families with multiple closely related members – including TCF/LEF, ETS, FOS/JUN, and SOX – HOMER lists a distinct consensus motif for each family member. To minimize clutter in *Figure 9C*, we have labeled only motifs from TF families with high statistical significance, and, within each TF family, we have labeled only the single motif with the highest statistical significance.

Motifs corresponding to the ETS family of transcription factors, which play a central role in EC development and function throughout the body (*Meadows et al., 2011*), were highly enriched in all of the samples for which there was a sufficient number of ATAC-seq peaks to generate statistically significant data. [As there were only 84 regions of accessible chromatin that were specific to posterior pituitary ECs, the P-values for motif enrichment cluster at the low end of the distribution for this sample (lower left plot in *Figure 9C*).] In WT cerebellum ECs but not in WT anterior pituitary ECs,

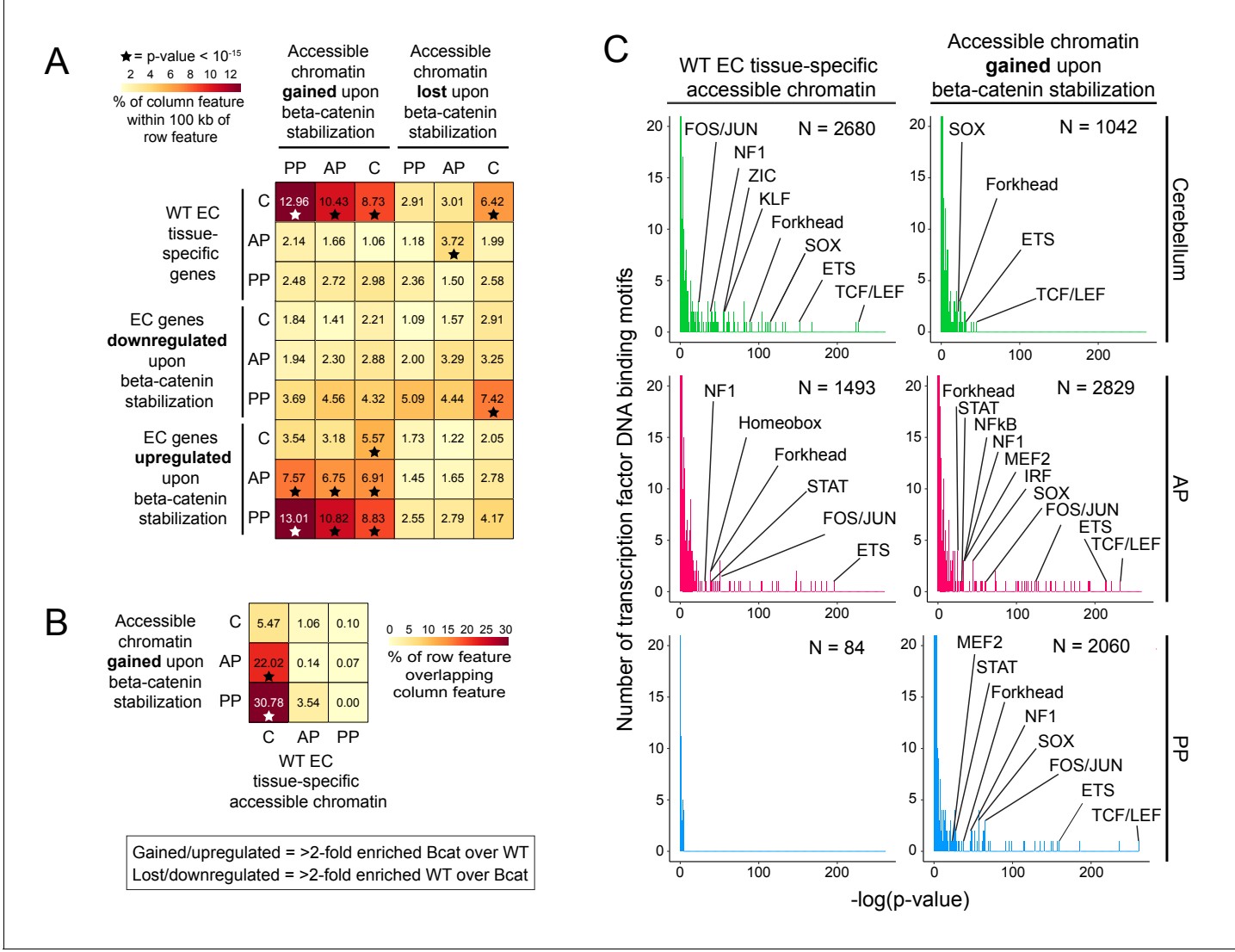

**Figure 9.** Regions of accessible chromatin that are induced by beta-catenin stabilization correlate with changes in EC gene expression and are highly enriched for TCF/LEF motifs. (**A**) Heatmap quantifying the number of accessible chromatin regions (ATAC-seq peaks) in FACS-purified ECs that were either gained or lost (i.e. ATAC-seq read counts changed >2 fold) upon beta-catenin stabilization and that are within 100 kb of (1) tissue-specific EC genes (as determined by comparisons between WT ECs) or (2) genes for which expression in ECs was either gained or lost upon beta-catenin stabilization. Statistical significance was calculated with a binomial test. (**B**) Heatmap indicating percentage overlap between accessible chromatin (ATAC-seq peaks) gained upon beta-catenin stabilization and accessible chromatin that is unique to each EC type (as determined by comparisons between WT ECs). There is significant overlap between accessible chromatin gained upon beta-catenin stabilization – especially in the two pituitary EC samples - and accessible chromatin unique to WT cerebellum ECs. Statistical significance calculated by Fisher's exact test. In (**A**) and (**B**), stars indicate statistical significance at p-value-$<10^{-15}$. (**C**) Transcription factor motif enrichment in accessible chromatin in WT vs. beta-catenin stabilized ECs. Histograms of the $-\log_{10}$(p-value) for 414 transcription factor DNA binding motifs that were tested for enrichment in EC tissue-specific accessible chromatin, based on comparisons of WT ECs (left column) and beta-catenin stabilized ECs (right column). The number of ATAC-seq peaks analyzed (i. e., peaks specific to EC tissue type and genotype) is indicated in each panel. The x-axis bin size is 1. The y-axis has been truncated. The vast majority of tested motifs have high p-values (i.e., low statistical significance) and thus are found in the first few bins. The statistically most significant TF motif families are labeled. Most TF families are represented by multiple closely related motifs, and only the motif with the lowest p-value for each family is labeled. C, cerebellum. AP, anterior pituitary. PP, posterior pituitary.

DOI: https://doi.org/10.7554/eLife.43257.025

the predominant TF motif that was enriched in accessible chromatin corresponds to the TCF/LEF family (*Figure 9C*, left), a finding that is in accord with our previous analyses of P7 brain ECs (*Sabbagh et al., 2018*). Strikingly, in all three samples of ECs with stabilized beta-catenin, TCF/LEF motifs showed the greatest enrichment in the newly gained accessible chromatin (*Figure 9C*, right).

In sum, this analysis shows that TCF/LEF motifs are highly enriched in genomic regions that are constitutively accessible in BBB-competent ECs (e.g. cerebellum ECs) or that become accessible in pituitary ECs upon activation of beta-catenin signaling. This finding lends strong support to a model in which beta-catenin regulates the BBB gene expression program via direct engagement of cis-regulatory elements adjacent to genes encoding BBB proteins (*Liebner et al., 2008*).

## Loss of WIF1 biases area postrema ECs to acquire a more BBB-like state

The experiments described up to this point have involved artificially activating beta-catenin signaling in ECs. If, as these experiments suggest, the high permeability state of CVO ECs requires low or absent beta-catenin signaling, then we might expect that the local environment created by CVO neurons and glia would be characterized by a low concentration of those canonical Wnt ligands that activate receptors on ECs and/or a high concentration of secreted Wnt inhibitors. Importantly, whatever mechanisms maintain the low level of beta-catenin signaling in CVO ECs must have a spatial distribution that closely matches that of the CVOs, since the BBB vs. non-BBB character of the vasculature switches abruptly at the border between each CVO and the adjacent CNS tissue, as seen in *Figure 10A* for the area postrema.

As a first step in exploring a possible role for local Wnt inhibition in maintaining vascular permeability in the CVOs and choroid plexus, we examined the in situ hybridization patterns obtained by the Allen Brain Institute (*Lein et al., 2007*) for secreted Wnt inhibitors (*Cruciat and Niehrs, 2013*). The two most striking patterns were those of *Wnt Inhibitory Factor-1* (*Wif1*; *Hsieh et al., 1999*), which is expressed in the area postrema (confirmed in *Figure 10B*), and *Sclerostin Domain Containing 1* (*Sostdc1*; *Ahn et al., 2010*), which is expressed in the choroid plexus. *Dickkopf-3* (*Dkk-3*) is expressed in the area postrema and also in the hippocampus and cortex. Our previous *in situ* hybridization analyses showed that *sFRP1* is specifically expressed in the choroid plexus and ciliary body (*Rattner et al., 1997*); the data from the Allen Brain Atlas confirms the choroid plexus expression and additionally shows no detectable *sFRP1* expression in the CVOs.

To assess the possible roles of SOSTDC1 and WIF1 in maintaining vascular permeability, *Sostdc1*$^{-/-}$ (*Ahn et al., 2010*), *Wif1*$^{-/-}$ (*Kansara et al., 2009*), and *Sostdc1*$^{-/-}$;*Wif1*$^{-/-}$ mice were examined by immunostaining the area postrema and choroid plexus for PLVAP, CLDN5, and GLUT1. All three genotypes were viable, and no histologic alterations could be ascribed to loss of *Sostdc1* (data not shown). However, loss of *Wif1* produced a ~2 fold increase in the fraction of vessels in the area postrema that express BBB markers [i.e. GLUT1+;PLVAP-; WT: $0.054 \pm 0.013$ (n = 10 mice); *Wif1*$^{-/-}$: $0.129 \pm 0.013$ (n = 13 mice); p=$1.46 \times 10^{-6}$; *Figure 10C and D*]. The simplest interpretation of these data is that local production of WIF1 in the area postrema reduces beta-catenin signaling in ECs, which contributes to the maintenance of a high permeability state. The modest phenotypic effect of the *Wif1* knockout implies that additional and partially redundant mechanisms of Wnt inhibition are likely to be operating.

## Discussion

The experiments reported here reveal a new aspect of vascular biology in the eye and brain: in contrast to the requirement for *elevated* EC beta-catenin signaling to maintain the BBB/BRB in most of the CNS, there is a requirement for *reduced* EC beta-catenin signaling to maintain the high permeability state of the CVO, choroid plexus, choriocapillaris, and ciliary body vasculatures. When the level of beta-catenin signaling is artificially elevated in ECs, high permeability brain and ocular vessels are converted to a low permeability BBB-like state. However, the data also indicate that beta-catenin signaling constitutes only part of the difference between BBB and non-BBB ECs, since the phenotypic conversion induced by beta-catenin signaling is incomplete. Presumably, other signals are also required to fully convert ECs to the BBB state.

The data presented here imply that highly permeable brain and ocular ECs retain substantial developmental plasticity. The ATAC-seq analyses represent one step in defining this plasticity at the

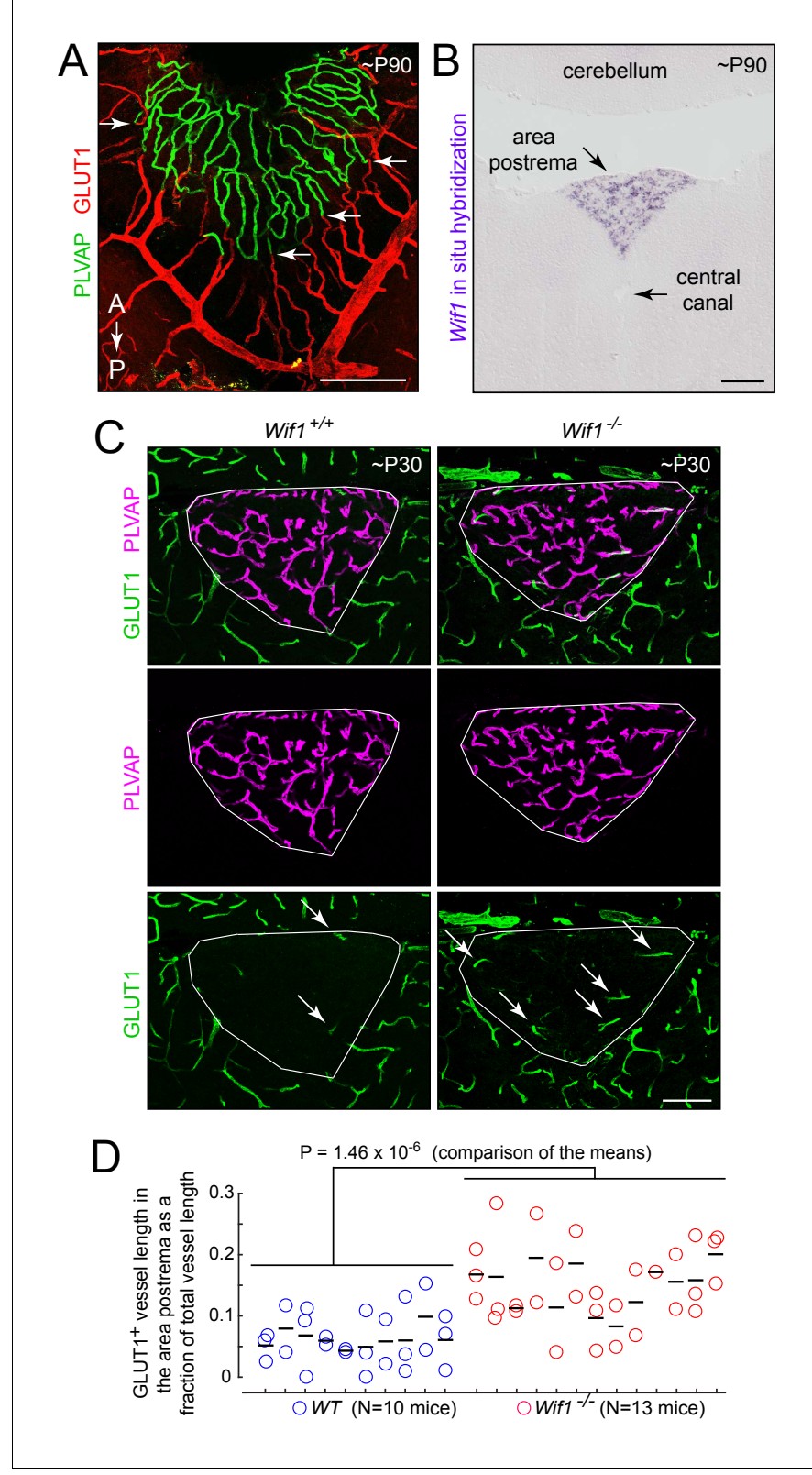

**Figure 10.** Loss of WIF1 increases the probability that ECs in the area postrema will assume a BBB-like state. (**A**) View of the dorsal surface of the area postrema showing a sharp transition at the junction between GLUT1-/PLVAP + ECs in the area postrema (center) and GLUT1+/PLVAP- ECs in the surrounding CNS tissue. Arrows mark transition points for four capillaries. A, anterior. P, posterior. (**B**) *In situ* hybridization with a *Wif1* probe to a coronal
*Figure 10 continued on next page*

*Figure 10 continued*

section through the brainstem (lower 60% of the image) and cerebellum (upper 20% of the image) at the level of the area postrema. *Wif1* transcripts (purple signal) are detected only in the area postrema. (C) Representative coronal sections through the area postrema in *Wif1*$^{+/+}$(i.e. WT) vs. *Wif1*$^{-/-}$littermates showing increased numbers of GLUT1+/PLVAP- ECs in the *Wif1*$^{-/-}$sample (arrows). The area postrema is demarcated by the smallest convex polygon that fully encompasses the GLUT1-/PLVAP+ vasculature. (D) Quantification of GLUT1+ blood vessel length as a fraction of the length of all vessels in the area postrema from WT (N = 10 mice) vs *Wif1*$^{-/-}$(N = 13 mice) littermates at P30. Vessels were manually traced from images like the ones shown in (C) and the result for each section is shown by a circle. Each vertical set of 1–3 circles represents the data from one mouse. Bars show mean ±S.D. The p-value was calculated from the mean for each mouse (black bars). Scale bars for (A) and (B), 200 μm. Scale bar for (C), 100 μm.

DOI: https://doi.org/10.7554/eLife.43257.026

level of the chromatin state, and – like the transcriptome changes that accompany beta-catenin stabilization – they show that changes in accessible chromatin do not fully recapitulate the BBB pattern, implying a requirement for additional signals or an irreversibility to some chromatin features of the high permeability EC gene expression program. The earlier observation that ectopic expression of Wnt7a in the mouse embryo – which presumably activates beta-catenin signaling in ECs – leads to GLUT1 accumulation in ECs in the peripheral vasculature (i.e. conversion to a BBB-like state; *Stenman et al., 2008*) suggested that plasticity in barrier gene expression may be a feature of immature ECs throughout the embryo. Our observation that posterior pituitary ECs show a greater response than anterior pituitary ECs to beta-catenin stabilization suggests that embryo-like plasticity is retained to a greater extent in highly permeable CNS ECs compared to non-CNS ECs.

For CNS ECs that reside within the neural retina and the non-CVO regions of the brain, the ligands Wnt7a, Wnt7b, and Norrin promote the BBB/BRB state. The observation that WIF1 is involved in maintaining the high permeability state of area postrema ECs implies that local inhibition of ligands that stimulate beta-catenin signaling is one mechanism by which the micro-environment can direct the development and maintenance of a highly permeable CNS vascular bed. It seems likely that the high permeability state depends on additional signals from CVO parenchymal cells, or, in the case of the choroid plexus, choriocapillaris, and ciliary body, from adjacent epithelial cells. For the choriocapillaris, current evidence suggests that VEGF may be one of the pro-permeability signals (*Tübingen Bevacizumab Study Group et al., 2007*; *Shimomura et al., 2009*). VEGF is also expressed in the CVOs, where it has been studied in the context of EC proliferation (*Morita et al., 2015*).

## Diversity of vascular structure and function in the CNS

It has long been appreciated that vascular structure and function varies between different organs and that this variability extends to sub-regions within organs, as seen, for example, in the differences in capillary EC structure in renal glomeri vs. the renal medulla (*Aird, 2007a*; *Aird, 2007b*; *Potente and Mäkinen, 2017*). The appreciation that the nervous system vasculature has distinctive permeability properties began with the discovery of the BBB in the late 19th and early 20th centuries (summarized in *Hawkins and Davis, 2005*) and was followed several decades later by the discovery that several small midline structures – now known to be the CVOs – lack a tight barrier (*Wislocki and Putnam, 1924*). More recent ultrastructural studies have revealed additional diversity within some CVOs. For example, neurons and glia within the area postrema, SFO, and VOLT are organized into distinct zones (*Price et al., 2008*; *Sisó et al., 2010*) and the microarchitecture of the capillaries differs between zones (*Gross, 1991*; *Gross, 1992*). Different levels of barrier function are also present in other CNS regions: the spinal cord has a modestly reduced version of the BBB (the blood-spinal cord barrier) and peripheral nerves have a further reduced version of the BBB (the blood-nerve barrier; *Weerasuriya and Mizisin, 2011*; *Reinhold and Rittner, 2017*).

In addition to permeability differences, capillary density within CVOs is several-fold higher than in other brain regions, such as the cerebral cortex, cerebellum, and brainstem. While capillary density in many parts of the body is regulated by oxygen demand (via VEGF signaling), this would seem to be an unlikely explanation for the high capillary density in CVOs, where high density is more likely a reflection of the requirement for efficient sampling of serum contents (in the sensory CVOs) or

efficient secretion of neuropeptides into the circulation (in the secretory CVOs). In contrast to the situation in the CVOs, the high density of the choriocapillaris, the extensive area of contact between the choriocapillaris and the adjacent RPE, and the high blood flow within the choroidal circulation are required to support the high glucose and oxygen consumption of rod and cone photoreceptors (*Yu and Cringle, 2001*; *Country, 2017*).

The functional and anatomic diversity within the CNS vasculature suggests that beta-catenin signaling is just one of multiple signaling pathways that play a role in locally shaping vascular structure and function. It will of interest to look for additional signaling molecules that vary locally and that could be candidates for mediating communication between parenchymal and vascular cells. It will also be of interest to identify additional examples of vascular diversity in different brain regions by comparing regional and single cell transcriptomes. Indeed, recent single cell RNA analyses have already provided evidence for previously unsuspected diversity among CNS ECs (*Vanlandewijck et al., 2018*; *Sabbagh et al., 2018*).

## Comparisons between brain and ocular vasculatures

The functional and structural similarities between the choroid plexus and ciliary body have long been noted. Both structures produce a plasma transudate – CSF and aqueous humor, respectively – and both consist of a fenestrated capillary plexus surrounded by epithelial cells that are connected by tight junctions, the latter creating the blood-CSF and blood-aqueous barriers. Similarly, the RPE, which is characterized by tight junctions and an absence of passive diffusion, is responsible for the blood-ocular barrier at the choriocapillaris-RPE interface. Similarities in the genetic programs in choroid plexus and ciliary body epithelial cells is suggested by the production of both choroid plexus and ciliary body tumors in mice with transgenic expression of the MYC oncoprotein in neural progenitors (*Shannon et al., 2018*). The experiments reported here show that the ciliary body, choriocapillaris, and choroid plexus ECs all share the property that they can be reprogrammed to a BBB-like state by activating beta-catenin signaling.

The epithelial-based blood-tissue barriers of the ciliary body, choriocapillaris/RPE, and choroid plexus contrast with the architecture of the BBB and BRB, where the blood-tissue barrier is at the level of the ECs. Barriers with an epithelial architecture are also found in reproductive organs. In the testis, developing germ cells are sequestered from permeable capillaries by a monolayer of Sertoli cells that are connected by tight junctions, thereby creating the blood-testis barrier (*Wen et al., 2018*); and in the placenta, fetal trophoblasts fuse to form a syncytium that separates the maternal and fetal circulatory systems (*Huppertz, 2008*).

## Clinical implications

Increases in CNS vascular permeability accompany a wide variety of neurologic conditions, including head trauma, multiple sclerosis, degenerative diseases (including Alzheimer disease), CNS infections, and a subset of brain tumors (*Obermeier et al., 2013*; *Zhao et al., 2015*). Among brain tumors, a dramatic example of increased vascular permeability is found in a subtype of medulloblastoma, referred to as WNT-medulloblastoma, which is characterized by a high level of beta-catenin signaling in the tumor cells. In mouse models of WNT-medulloblastoma, the intra-tumor capillaries resemble CVO capillaries in their high density, tortuosity, and GLUT1-/PLVAP+ phenotype (*Phoenix et al., 2016*). In these models, the high level of tumor cell-specific beta-catenin signaling is accompanied by high-level production of Wnt inhibitors DKK1 and WIF1, which reprogram the intra-tumor vascular phenotype by inhibiting beta-catenin signaling in ECs. In contrast, the intra-tumor capillaries in other medulloblastoma subtypes resemble conventional BBB-type CNS capillaries. High vascular permeability enhances sensitivity to chemotherapy and likely accounts for the better prognosis of WNT-medulloblastomas compared to other medulloblastoma subtypes. These observations suggest that therapeutic strategies aimed at inhibiting beta-catenin signaling within tumor vasculature could enhance responsiveness to chemotherapy in a wide variety of CNS tumors. In the context of the work presented here, the similarities between capillaries in WNT-medulloblastomas and in CVOs imply that a detailed understanding of the mechanisms responsible for generating and maintaining high permeability CVO capillaries could inform strategies for enhancing vascular permeability to improve CNS drug delivery.

The study of CVO capillary permeability – and, in particular, the observation that it is reduced in response to activated beta-catenin signaling – could also inform strategies aimed at reducing CNS vascular permeability in other pathologic contexts. A common attribute of many neurologic disorders that feature loss of BBB integrity is an increase in CNS inflammation (*Liebner et al., 2018*). One hypothesis to explain disease-associated vascular permeability is that CNS inflammation recapitulates the permeability-promoting mechanisms that are active in the CVOs. As previous work has shown that the BBB state remains plastic throughout life and that its maintenance requires continuous beta-catenin signaling (*Liebner et al., 2008*; *Wang et al., 2012*; *Zhou et al., 2014*), a change in the local tissue environment that directly or indirectly antagonizes beta-catenin signaling – for example, by activating an antagonistic signaling pathway in ECs or inhibiting/degrading Wnt ligands or receptors – could lead to a loss of BBB competence. This line of reasoning suggests that enhancing beta-catenin signaling in CNS ECs could be a treatment strategy for pathological BBB breakdown.

## The role of beta-catenin signaling in CNS vascular biology

As noted in the preceding paragraph, the BBB/BRB state remains plastic throughout life. The present work extends this observation by showing that the high permeability ECs in the CVO, choroid plexus, choriocapillaris, or ciliary body are similarly plastic. Interestingly, the transcriptome comparisons between cerebellar and pituitary ECs show that, with elevated beta-catenin signaling, cerebellar ECs reduce the levels of pituitary EC-specific transcripts even further than their normally low levels. This suggests that the gene expression set point of the standard BBB state is not at the extreme end of the BBB/non-BBB continuum.

A striking aspect of adult CNS EC plasticity, is the quantized phenotypic conversion of individual ECs in response to changes in beta-catenin signaling. This behavior suggests the existence of positive feedback. While some proteins that accumulate in response to beta-catenin signaling are likely involved in negative feedback regulation – including, for example, the induction of Axin2, which is part of the beta-catenin destruction complex – the accumulation of LEF1, which cooperates with beta-catenin as a heteromeric transcriptional regulator of target genes, is likely involved in positive feedback. Positive feedback in beta-catenin signaling could facilitate the development and maintenance of spatially precise and temporally stable vascular territories that differ in their barrier properties, as seen, for example, in the sharp junctions between CVO and non-CVO vascular territories.

## Materials and methods

**Key resources table**

| Reagent type (species) or resource | Designation | Source or reference | Identifiers | Additional information |
|---|---|---|---|---|
| Genetic reagent (*Mus musculus*) | *Ctnnb1^flex3^* | *Harada et al., 1999* | | |
| Genetic reagent (*M. musculus*) | *Ndp^-^* | The Jackson Laboratory | Stock No: 012287; RRID:IMSR_JAX:012287 | |
| Genetic reagent (*M. musculus*) | *Fz4^-^* | The Jackson Laboratory | Stock No: 012823; RRID:IMSR_JAX:012823 | |
| Genetic reagent (*M. musculus*) | *Fz4^CKO^* | The Jackson Laboratory | Stock No: 011078; RRID:IMSR_JAX:011078 | |
| Genetic reagent (*M. musculus*) | *Tie2-Cre* | The Jackson Laboratory | Stock No: 008863; RRID:IMSR_JAX:008863 | |
| Genetic reagent (*M. musculus*) | *Tie2-GFP* | The Jackson Laboratory | Stock No: 003658; RRID:IMSR_JAX:003658 | |
| Genetic reagent (*M. musculus*) | *Pdgfb-CreER* | *Claxton et al., 2008* | | |
| Genetic reagent (*M. musculus*) | *Wif1^-^* | *Kansara et al., 2009* | | |
| Genetic reagent (*M. musculus*) | *Sostdc1^-^* | *Ahn et al., 2010* | | |

*Continued on next page*

*Continued*

| Reagent type (species) or resource | Designation | Source or reference | Identifiers | Additional information |
|---|---|---|---|---|
| Genetic reagent (*M. musculus*) | *Rpl22ribotag* | The Jackson Laboratory | Stock No: 029977; RRID:IMSR_JAX:029977 | |
| Genetic reagent (*M. musculus*) | *R26-LSL-tdTomato -2A-H2B-GFP* | The Jackson Laboratory | Stock No: 030867; RRID:IMSR_JAX:030867 | |
| Antibody | anti-GLUT1 (rabbit polyclonal) | Thermo Fisher Scientific | Cat no: RB-9052-P1; RRID:AB_177895 | |
| Antibody | anti-mouse PLVAP/MECA-32 (rat monoclonal) | BD Biosciences | Cat no: 553849; RRID:AB_395086 | |
| Antibody | anti-CLDN5 Alexa Fluor 488 conjugate (mouse monoclonal) | Thermo Fisher Scientific | Cat no: 352588; RRID:AB_2532189 | |
| Antibody | anti-GFP Alexa Fluor 488 conjugate (rabbit polyclonal) | Thermo Fisher Scientific | Cat no: A21311; RRID:AB_221477 | |
| Antibody | anti-LEF1 (rabbit monoclonal) | Cell Signaling Technologies | Cat no: C12A5; RRID:AB_823558 | |
| Antibody | anti-Occludin (rabbit polyclonal) | Invitrogen | Cat no: 406100; RRID:AB_2533473 | |
| Antibody | anti-ZO1 (rabbit polyclonal) | Invitrogen | Cat no: 40–2200; RRID:AB_2533456 | |
| Antibody | anti-MDR1 (rabbit monoclonal) | Cell Signaling Technologies | Cat no: 13978S | |
| Antibody | anti-Vimentin (chicken polyclonal) | EMD Millipore | Cat no: AB5733; RRID:AB_11212377 | |
| Peptide, recombinant protein | Tn5 transposase | Illumina | Cat no: FC-121–1030 | |
| Commercial assay or kit | Worthington Papain Dissociation Kit | Worthington Biochemical Corporation | Cat no: LK003160 | |
| Chemical compound, drug | 4-hydroxytamoxifen | Sigma-Aldrich | Cat no: H7904 | |
| Software, algorithm | Salmon | *Patro et al., 2017* | | |
| Software, algorithm | Kallisto | *Bray et al., 2016* | RRID:SCR_016582 | |
| Software, algorithm | deepTools | *Ramírez et al., 2016* | RRID:SCR_016366 | |
| Software, algorithm | tximport | *Soneson et al., 2016* | RRID:SCR_016752 | |
| Software, algorithm | EBSeq | *Leng et al., 2015* | RRID:SCR_003526 | |
| software, algorithm | MACS2 | *Zhang et al., 2008* | | |
| Software, algorithm | DiffBind | *Ross-Innes et al., 2012* | RRID:SCR_012918 | |
| Software, algorithm | Bedtools | *Quinlan and Hall, 2010* | RRID:SCR_006646 | |
| Other | Texas Red streptavidin | Vector Laboratories | Cat no: SA-5006; RRID:AB_2336754 | |
| Other | Sulfo-NHS-biotin | Thermo Fisher Scientific | Cat no: 21217 | |

## Mice

The following mouse alleles were used: *Ctnnb1flex3* (*Harada et al., 1999*); *Ndp-* (*Ye et al., 2009*; JAX 012287), *Fz4-*(*Xu et al., 2004*; JAX 012823); *Fz4CKO*(*Ye et al., 2009*; JAX 011078); *Tie2-Cre* (also known as *Tek-Cre*; *Kisanuki et al., 2001*; JAX 008863); *Tie2-GFP* mice (also known as *Tek-GFP*; *Motoike et al., 2000*; JAX 003658); *Pdgfb-CreER* (*Claxton et al., 2008*); *Wif1-*(*Kansara et al., 2009*); *Sostdc1-*(*Ahn et al., 2010*); *Rpl22ribotag*(*Sanz et al., 2009*; JAX 029977) and *R26-LSL-tdTo-mato-2A-H2B-GFP* (*Wang et al., 2018*; JAX 030867). All mice were housed and handled according

to the approved Institutional Animal Care and Use Committee (IACUC) protocol MO16M367 of the Johns Hopkins Medical Institutions.

## Antibodies and other reagents

The following antibodies were used for tissue immunohistochemistry: rabbit anti-GLUT1 (Thermo Fisher Scientific RB-9052-P1); rat anti-mouse PLVAP/MECA-32 (BD Biosciences 553849); mouse anti-CLDN5, Alexa Fluor 488 conjugate (Thermo Fisher Scientific 352588); rabbit anti-GFP, Alexa Fluor 488 conjugate (Thermo Fisher Scientific A21311); rabbit anti-6xMyc (JH6204), rabbit mAb anti-LEF1 (Cell Signaling Technologies C12A5), rabbit anti-ZO-1 (Invitrogen 40–2200), rabbit anti-Occludin (Invitrogen 406100), rabbit mAb anti-MDR1 (E1Y7S; Cell Signaling Technology 13978S), chicken anti-Vimentin (EMD Millipore Corp AB5733), and rabbit anti-MFSD2A (a kind gift of David Silver, Duke-NUS Medical School). Alexa Fluor-labeled secondary antibodies and GS Lectin (Isolectin GS-IB4) were from Thermo Fisher Scientific. Texas Red streptavidin was from Vector Laboratories (SA-5006). Sulfo-NHS-biotin was from Thermo Fisher Scientific (catalogue #21217).

## Tissue processing and immunohistochemistry

Tissue were prepared and processed for immunohistochemical analysis as described by *Wang et al. (2012)* and *Zhou et al. (2014)*. In brief, mice were deeply anesthetized with ketamine and xylazine and then perfused via the cardiac route with 1% PFA in phosphate buffered saline (PBS) followed by 100% cold methanol dehydration overnight at 4°C. For PFA-sensitive antigens (MDR1, OCLN, and ZO-1), mice were deeply anesthetized with ketamine and xylazine and then perfused via the cardiac route with PBS followed by 100 mls of cold methanol, and then the brains and pituitaries were additionally fixed by immersion in 100% cold methanol overnight at 4°C. Tissues were re-hydrated the following day in 1x PBS at 4°C for at least 3 hr before embedding in 3% agarose. Tissue sections of 100–180 μm thickness were cut using a vibratome (Leica).

For the vascular permeability analysis, mice were injected intraperitoneally with Sulfo-NHS-biotin (200 μl of 20 mg/ml Sulfo-NHS-biotin in PBS) ten minutes prior to intracardiac perfusion. Covalently bound biotin was visualized in vibratome sections with Streptavidin.

Sections were incubated overnight with primary antibodies, secondary antibodies, or Texas Red streptavidin diluted in 1x PBSTC (1x PBS + 1% Triton X-100 +0.1 mM $CaCl_2$)+10% normal goat serum (NGS). Incubation and washing steps were performed at 4°C. Sections were washed at least 3 times with 1x PBSTC over the course of 6 hr, and subsequently incubated overnight with secondary antibodies diluted in 1x PBSTC +10% NGS. If a primary antibody raised in rat was used, secondary antibodies were additionally incubated with 1% normal mouse serum (NMS) as a blocking agent. The next day, sections were washed at least 3 times with 1x PBSTC over the course of 6 hr, and flat-mounted using Fluoromount G (EM Sciences 17984–25). Sections were imaged using a Zeiss LSM700 confocal microscope, and processed with ImageJ, Adobe Photoshop, and Adobe Illustrator software.

## *In situ* hybridization (ISH)

ISH on tissue sections was performed essentially as described by *Schaeren-Wiemers and Gerfin-Moser (1993)*, using an alkaline phosphatase conjugated anti-digoxigenin antibody and NBT/BCIP histochemistry to visualize hybridization. Digoxigenin-labeled riboprobes were synthesized with T7 RNA polymerase from mouse *Wif1* cDNA (nucleotides 1225–2147 in the numbering system in which the coding region spans nucleotides 340–1479, NCBI accession number NM_011915). Images were captured on a Zeiss Imager Z1 microscope using Zeiss AxioVision 4.6 software.

## 4HT preparation and administration

Solid 4HT (Sigma-Aldrich H7904) was dissolved at 20 mg/ml in ethanol by extensive vortexing. Sunflower seed oil (Sigma-Aldrich S5007) was added to dilute the 4HT to 2 mg/ml and aliquots were stored at −80°C. Thawed aliquots were diluted with Sunflower seed oil to a final concentration of 1 mg/ml 4HT. All injections were performed intraperitoneally.

## Quantification of vasculature in the area postrema

For quantifying relative vascular density in the area postrema, 100 µm thick coronal sections from *Wif1*$^{+/+}$ or *Wif1*$^{-/-}$ littermates were stained for GLUT1 and PLVAP, and then stained with DAPI. The area postrema is ~500 um from front to back and was typically imaged from 2 to 3 vibratome sections. Confocal images were scanned at 10–12 µm intervals along the Z-axis, of which four images were Z-stacked. Starting with each Z-stacked image, the perimeter of the area postrema was delineated with the smallest convex polygon that encompassed all of the PLVAP+ ECs, as shown in *Figure 10C*. All of the blood vessels within the designated polygon were manually traced using Adobe Illustrator software. The lengths of the PLVAP+ traced vessels and the lengths of the GLUT1+ vessels were separately quantified by calculating pixel coverage as a fraction of the total area using ImageJ. The length of GLUT1+ vessels was then divided by the sum of the lengths of GLUT1+ and PLVAP+ vessels for the vasculature within each area postrema polygon.

The R software package was used to generate plots and to perform statistical analyses. The mean ± standard deviation is shown. Statistical significance was determined by the unpaired t-test.

## Transmission electron microscopy (TEM) and quantification of fenestrae

For TEM of brain and eye tissues, P10 mice [control (*Ctnnb1*$^{flex3/+}$) or experimental (*Ctnnb1*$^{flex3/+}$; *Pdgfb-CreER*)] were given a single dose of 400 ug of 4HT intraperitoneally and sacrificed at ~P40. Following intra-cardiac perfusion in Karnovsky's fixative (2% glutaraldehyde, 2% paraformaldehyde, 1.5 mM $CaCl_2$, and 1.5 mM $MgCl_2$ in PBS, pH 7.2), tissues of interest were dissected and immersion-fixed in Karnovsky's fixative overnight at 4°C. Tissues were rinsed with 0.05 N cacodylate buffer supplemented with 1.5 mM $CaCl_2$ and 1.5 mM $MgCl_2$, and fixed in 1% $OsO_4$ on ice. Tissues were then rinsed with water, dehydrated, and embedded in Epoxy resin. Sections were cut with a diamond knife, placed on copper grids, stained with uranyl acetate in methanol (filtered twice through a 0.22 um filter) and viewed with a Hitachi transmission electron microscope.

Morphometry was carried out in ImageJ. Fenestrae were readily seen at 15,000x magnification (corresponding to 0.2069 pixels/nm). The perimeters of the blood capillaries were manually traced and the length of the traces measured in ImageJ. The fenestrae were counted using the Cell Counter plugin, and the number of fenestrae per 100 µm was calculated.

## Immunoprecipitation of RiboTag-labelled polyribosomes

6–8 week old WT control and beta-catenin stabilized mice were used for the RiboTag experiments. Immunoprecipitation of the RiboTag-labelled polyribosomes were carried out as previously described (*Sanz et al., 2009*). Cortex and choroid plexus were promptly dissected following cervical dislocation and the tissues snap-frozen. The choroid plexi from 3 to 4 animals were pooled as one sample. For immunoprecipitation, tissues were homogenized in 500 µl (choroid plexus) or 700 µl (cerebral cortex) polysome buffer [50 mM Tris, pH 7.5, 250 mM sucrose, 100 mM KCl, 12 mM $MgCl_2$, 1% Nonidet P-40, 1 mM DTT, 200 U/ml RNasin Plus (Promega), 1 mg/ml heparin, 100 g/ml cycloheximide, 0.5 mM spermidine, and complete EDTA-free protease inhibitor cocktail (Roche)]. Homogenates were centrifuged at 10,000xg for 10 min at 4°C to create a postmitochondrial supernatant. 25 µl and 40 µl aliquots of choroid plexus homogenate and cortex homogenate, respectively, were saved as input samples. Four µl of mouse monoclonal anti-HA antibody (HA.11, ascites fluid; Covance) were added to the supernatants and rotated for 4 hr at 4°C. 100 µl protein G magnetic beads (Dynabeads; Invitrogen) were washed twice with polysome buffer and added directly to the antibody-coupled polysomes and rotated overnight at 4°C. The following day, samples were placed in a magnet on ice and supernatants recovered before washing the pellets 4 × 5 min in high salt buffer (50 mM Tris, pH 7.5, 300 mM KCl, 12 mM MgCl 2, 1% Nonidet P-40, 1 mM DTT, 20 µg/ml cycloheximide). To prepare total RNA, 350 µl of Qiagen RLT buffer was added to the beads or to the input samples. Total RNA was prepared according to manufacturer's instructions using the RNeasy Plus Micro kit (Qiagen), and the quantity and quality of the RNA were assessed using a Bioanalyzer (Agilent).

## Tissue dissection, EC purification, and RNA and DNA sample preparation

To control for the possibility of sex-dependent differences, male mice were used for RNA-seq and ATAC-seq. Viable ECs were isolated using the Worthington Papain Dissociation System (LK003160, Worthington Biochemical Corporation, Lakewood, NJ) and a MoFlo XDP Sorter (Beckman Coulter, Brea, CA) as previously described (*Sabbagh et al., 2018*), with propidium iodide negative cells considered as viable. For pituitaries, 20–30 mice between P30 and P90 were used for each independent replicate. The mice consisted of control WT mice (*Tie2-GFP*) and EC-specific beta-catenin stabilized (*Ctnnb1^{flex3/+}*;*Pdgfb-CreER;Tie2-GFP* that had received 100 ug 4HT at P10). Pituitaries were dissected from the base of the skull and place in Dulbecco's PBS (DPBS). Posterior pituitaries (a small whitish tissue with a clear border occupying the dorso-medial pituitary) was gently separated from the anterior pituitary and the 20–30 anterior and posterior pituitaries were collected in separate tubes for subsequent papain dissociation. RNA was extracted from GFP-positive and GFP-negative cells that were FACS sorted directly into QIAGEN Buffer RLT Plus and then processed using the RNeasy Micro Plus kit (74034, QIAGEN, Venlo, Netherlands). For ATAC-seq,~50,000 GFP-positive FACS-sorted cells were gently centrifuged and then resuspended in ice-cold lysis buffer (0.25 M sucrose, 25 mM KCl, 5 mM MgCl₂, 20 mM Tricine-KOH, 0.1% Igepal CA-630) and immediately centrifuged at 500 x g for 10 min at 4°C to prepare nuclei. The resulting nuclear pellet was resuspended in a 50 ul reaction volume in Tn5 transposase and transposase reaction buffer (FC-121–1030, Illumina Inc, San Diego, CA), and the tagmentation reaction was incubated at 37°C for 30 min.

## Library preparation and sequencing

Each RNA-seq and ATAC-seq analysis was conducted on two biological replicates. Libraries for RNA-seq and ATAC-seq were prepared as previously described (*Buenrostro et al., 2015*; *Sabbagh et al., 2018*). For RNA-seq, total RNA was converted to cDNA and amplified (Ovation Ultralow System V2-32, 0342HV, NuGEN Technologies). Amplified cDNA was fragmented, end-repaired, linker-adapted, and single-end sequenced for 75 cycles on a NextSeq500 (Illumina). Tagmented DNA was purified using QIAGEN MinElute Gel Extraction kit (28604, Qiagen). ATAC-seq libraries were PCR amplified for 11 cycles. Agencourt AMPure XP beads (A63880, Beckman Coulter) were used to purify ATAC-seq libraries, which were then paired-end sequenced for 36 cycles on a NextSeq500 (Illumina).

## Data analysis

Most data analyses were performed as previously described (*Sabbagh et al., 2018*). For basic data processing, exploration, and visualization, we used deepTools (*Ramírez et al., 2016*), BEDTools (*Quinlan and Hall, 2010*), RStudio, the tidyverse collection of R packages (*Wickham, 2017*), ggplot2 (*Wickham, 2009*), and pheatmap (*Kolde, 2015*). Reads were aligned to the mm10 genome using Bowtie2 (*Langmead and Salzberg, 2012*).

## RNA-seq data analysis

Salmon version 0.10.2 (*Patro et al., 2017*) was used to quantify expression of transcripts from RNA-seq experiments using mm10/GRCm38_92 Ensembl transcriptome (salmon quant -l A -g Mus_musculus.GRCm38.92.chr.gtf.gz –validateMappings –rangeFactorizationBins 4 –incompatPrior 0.0 –useVBOpt –seqBias –gcBias –posBias –biasSpeedSamp 10). To visualize RNA-seq data on an IGV browser (*Robinson et al., 2011*; *Thorvaldsdóttir et al., 2013*), kallisto version 0.44.0 (*Bray et al., 2016*) was used to generate alignment bam files (kallisto quant –bias –single -l 250 s 25 –genomebam -g Mus_musculus.GRCm38.92.chr.gtf.gz -c mouse.mm10.genome) and then deepTools was used to generate bigwig files (bamCoverage -bs 1 –normalizeUsing RPKM).

To convert transcript-level abundances to the gene-level for further downstream analyses, we used tximport (*Soneson et al., 2015*). Differentially expressed genes were identified using EBSeq version 1.20.0 (*Leng and Kendziorski, 2015*). To filter out background transcripts from surrounding parenchymal cells, a set of EC-enriched transcripts was determined for each tissue by comparing RNA data from GFP-positive and GFP-negative sorted cells. A transcript was considered EC-enriched if it met the following three criteria: (1) a minimum two-fold enrichment in GFP-positive compared to GFP-negative samples; (2) a posterior probability of differential expression (PPDE)

greater than or equal to 0.95 [PPDE = (1 - false discovery rate)], that is an FDR < 0.05; (3) relative expression greater than or equal to 10 transcripts per million (TPM) in both biological replicates. A gene was considered to be differentially expressed between EC subtypes if it met the following criteria: (1) EC-enriched with minimum of two-fold enrichment between one subtype and both other subtypes; (2) a PPDE greater than or equal to 0.95; (3) a TPM value greater than or equal to 10 in both biological replicates. Principal component analysis was performed on 'regularized' log-transformed data using the DESeq2 *rlog* and *plotPCA* function (*Love et al., 2014*). For RiboTag samples, EC-enriched transcripts were determined by comparing the pull-down sample against the input sample.

## ATAC-seq data analysis

ATAC-seq data were aligned using Bowtie2 (Version 2.3.2 t -X 2000 –no-mixed –no-discordant) and then duplicate reads were removed (*picard MarkDuplicates*). Peaks were called using MACS2 (Version 2.1.1.20160309 *callpeak* –nomodel –keep-dup all –shift −100 –extsize 200 –call-summits) (*Zhang et al., 2008*). Peaks were then filtered for fold-change >2 and -log(qvalue)>2. deepTools was used to visualize ATAC-seq peaks on the browser (bamCoverage -bs 1 –normalizeUsing RPKM). To identify differential ATAC-seq peaks between ECs isolated from adult cerebellum, anterior pituitary, and posterior pituitary, DiffBind was used (*Stark and Brown, 2011*; *Ross-Innes et al., 2012*) with EdgeR (*Robinson et al., 2010*). For each pairwise comparison, DiffBind was used to develop a set of consensus peaks between replicates using the requirement that peaks must be in both replicates (minOverlap = 2). To retrieve a set of high-confidence, cell type-enriched peaks, we filtered for peaks with an absolute fold difference >2 and FDR < 0.05. Principal component analysis was performed on 'regularized' log-transformed data using the DESeq2 *rlog* and *plotPCA* function. To generate scatter plots of ATAC-seq data, the union of consensus peaks for each WT pairwise comparison was used with *dba.count* to generate normalized counts at each peak for each sample. To identify transcription factor DNA binding motifs enriched in ATAC-seq peaks, the HOMER suite of tools was used for motif discovery (*Heinz et al., 2010*), in particular, *findMotifsGenome.pl* (-size given).

## Feature overlap analysis

*Bedtools intersect* (-u) was used to determine features that overlapped with other features by greater than or equal to 1 bp. *Bedtools window* (-w 100000) was used to identify ATAC-seq peaks within 100 kb of various sets of genes. For *Figure 9A*, statistically significant enrichment was tested by binomial test using the R command *binom.test(overlap, total number of column feature, p=total base pairs of genome covered by 100 kb window around genes in row feature/total base pairs of mm10 genome,alternative="greater')*. For *Figure 9B*, statistically significant enrichment was tested by Fisher's exact test using the R command *fisher.test(matrix(c(overlap, total number of row feature – overlap, total number of column feature – overlap, discrete called peaks – total number of row feature – total number of column feature), nrow = 2), alternative="greater')*.

## Data access

Data files are available at GEO accession GSE122117. Processed RNA-seq data can be accessed at the VECTRDB website (https://markfsabbagh.shinyapps.io/vectrdb/).

## Acknowledgments

The authors would like to thank Dr. Maketo Taketo (Kyoto University) for the *Ctnnb1^flex3^* mice, Alex Grinberg (NICHD, NIH) for the *Wif1⁻* mice, Robb Krumlauf (Stowers Institute) for *Sostdc1⁻* mice, David Silver (Duke-NUS Medical School) for rabbit anti-MFSD2A antiserum, Haiping Hao and Linda Orzolek (Johns Hopkins University Deep Sequencing and Microarray Core Facility) for RNA library preparation and for NextGen sequencing, Michael Delannoy (Johns Hopkins Microscope Core Facility) for assistance with TEM; and Tao-Hsin Chang, Chris Cho, and Jacob Heng for helpful discussions and/or comments on the manuscript. This work was supported by the Howard Hughes Medical Institute, the National Eye Institute (R01 EY018637), and the Arnold and Mabel Beckman Foundation.

## Additional information

### Competing interests

Jeremy Nathans: Reviewing editor, *eLife*. The other authors declare that no competing interests exist.

### Funding

| Funder | Grant reference number | Author |
|--------|------------------------|--------|
| Howard Hughes Medical Institute | | Yanshu Wang<br>Xiaowu Gu<br>John Williams<br>Jeremy Nathans |
| National Eye Institute | R01 EY018637 | Amir Rattner<br>Jeremy Nathans |
| Eunice Kennedy Shriver National Institute of Child Health and Human Development | | Mark F Sabbagh |
| Arnold and Mabel Beckman Foundation | | Yanshu Wang<br>Amir Rattner |

The funders had no role in study design, data collection and interpretation, or the decision to submit the work for publication.

### Author contributions

Yanshu Wang, Conceptualization, Validation, Investigation, Visualization, Writing—original draft, Writing—review and editing; Mark F Sabbagh, Data curation, Software, Formal analysis, Funding acquisition, Investigation, Visualization, Writing—original draft, Writing—review and editing; Xiaowu Gu, Conceptualization, Formal analysis, Investigation, Methodology; Amir Rattner, Formal analysis, Investigation; John Williams, Investigation; Jeremy Nathans, Conceptualization, Formal analysis, Supervision, Funding acquisition, Methodology, Writing—original draft, Project administration, Writing—review and editing

### Author ORCIDs

Mark F Sabbagh (iD) http://orcid.org/0000-0003-1996-5251
Xiaowu Gu (iD) http://orcid.org/0000-0003-2266-5516
Amir Rattner (iD) http://orcid.org/0000-0001-9542-6212
Jeremy Nathans (iD) http://orcid.org/0000-0001-8106-5460

### Ethics

Animal experimentation: This study was performed in strict accordance with the recommendations in the Guide for the Care and Use of Laboratory Animals of the National Institutes of Health. All of the animals were handled according to the approved Institutional Animal Care and Use Committee (IACUC) protocol MO16M367 of the Johns Hopkins Medical Institutions.

### Decision letter and Author response

Decision letter https://doi.org/10.7554/eLife.43257.034
Author response https://doi.org/10.7554/eLife.43257.035

## Additional files

### Supplementary files

• Supplementary file 1. Gene expression data. (A-B) Transcript abundances in raw counts (A) and TPMs (B). (C-F) Genes differentially enriched in WT cerebellum ECs (C), either anterior or posterior pituitary ECs (D), WT anterior pituitary ECs (E), or WT posterior pituitary ECs (F). (G-H) Genes

upregulated by beta-catenin stabilization in either anterior (G) or posterior (H) pituitary ECs that overlap the genes in (C).
DOI: https://doi.org/10.7554/eLife.43257.027

• Supplementary file 2. Ribotag gene expression data. (A-B) Transcript abundances in raw counts (A) and TPMs (B). (C-D) Genes differentially enriched in WT cortical ECs (C) or WT choroid plexus ECs (D). (E) Genes upregulated by beta-catenin stabilization in choroid plexus ECs that overlap the genes in (C).
DOI: https://doi.org/10.7554/eLife.43257.028

• Supplementary file 3. Accessible chromatin peaks in each EC subtype. (A-F) ATAC-seq peaks for each EC subtype called using the full range of ATAC-seq fragment lengths. (G-I) Differential ATAC-seq peaks for WT cerebellum ECs (G), WT anterior pituitary ECs (H), and WT posterior pituitary ECs (I) in either pairwise comparison. (J-L) Unique differential ATAC-seq peaks for WT cerebellum ECs (J), WT anterior pituitary ECs (K), and WT posterior pituitary ECs (L) in either pairwise comparison. (M-O) ATAC-seq peaks gained upon beta-catenin stabilization in WT cerebellum ECs (M), WT anterior pituitary ECs (N), and WT posterior pituitary ECs (O). (P-R) ATAC-seq peaks lost upon beta-catenin stabilization in WT cerebellum ECs (P), WT anterior pituitary ECs (Q), and WT posterior pituitary ECs (R). (S-T) ATAC-peaks gained upon beta-catenin stabilization in either anterior (S) or posterior (T) pituitary ECs that overlap the peaks in (J).
DOI: https://doi.org/10.7554/eLife.43257.029

• Transparent reporting form
DOI: https://doi.org/10.7554/eLife.43257.030

## Data availability

Sequencing data have been deposited in GEO under accession code GSE122117.

The following dataset was generated:

| Author(s) | Year | Dataset title | Dataset URL | Database and Identifier |
|---|---|---|---|---|
| Wang Y | 2019 | The role of beta-catenin signaling in regulating barrier vs. non-barrier gene expression programs in circumventricular organ and ocular vasculatures | https://www.ncbi.nlm.nih.gov/geo/query/acc.cgi?acc=GSE122117 | NCBI Gene Expression Omnibus, GSE122117 |

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
