## [Decision Letter]

Thank you for submitting your article "Beta-catenin signaling affects barrier vs non-barrier gene expression in circumventricular organ and ocular vasculatures" for consideration by *eLife*. Your article has been reviewed by three peer reviewers, including Elisabetta Dejana as the Reviewing Editor and Reviewer #1, and the evaluation has been overseen by Didier Stainier as the Senior Editor. The following individuals involved in review of your submission have agreed to reveal their identity: Dritan Agalliu (Reviewer #2); Christer Betsholtz (Reviewer #3).

The reviewers have discussed the reviews with one another and the Reviewing Editor has drafted this decision to help you prepare a revised submission.

Summary and essential revisions:

Reviewers agreed that the study is interesting and the major conclusions of the paper, once substantiated by additional data as outlined in the reviews, would in principle be sufficient to warrant publication in *eLife*. The two major aspects to revise are:

1) Better characterization of the permeability properties of the CVO endothelial cells;

2) A more precise characterization of the composition of endothelial cell junctions in the different conditions.

Reviewer #1:

In the present paper Wang et al. study the role of canonical Wnt /beta-catenin signaling on the permeability properties of the capillaries of the circumventricular organs and choriocapillaris. These vessels present different anatomical and functional characteristics that strongly influence their low control of permeability. Applying excellent imaging analysis and a large set of murine strains of genetically modified mice the authors bring evidence that low beta-catenin signaling is responsible, at least to a significant extent, for low permeability control. This observation is intriguing since the same signaling pathway appears to be required for correct differentiation of the blood brain barrier but should be strongly reduced in the CVO vessels to favor low permeability and dynamic exchanges. This work also shows that brain endothelial cells maintain a certain degree of plasticity in the adult being capable to positively or negatively respond to Wnt signaling.

The paper is well done and certainly of interest but some aspects should be better defined to make it more complete:

1) Along the manuscript the way to define the blood brain barrier is mostly based on upregulation of claudin-5 expression, increase in GLUT-1 and decreased PLVAP. This may be an oversimplification also considering the extreme variety of the vasculature in the brain or CVO capillaries. I strongly recommend adding staining of other tight junction elements such as cingulin or ZOs and possibly of specific transporters, at least in some key figures.

2) The analysis of chromatin accessibility and RNA confirm the fact that beta-catenin is a major player in the maintenance of permeability control in the different regions of the brain. However, beta-catenin concentration should be tightly controlled to avoid pathological reactions. In CVO, as reported in this paper, optimal level of beta-catenin activation is due to the fine balance between activators and inhibitors such as WIF-1 and *Sostdc1*. These observations are novel and of interest. However, overall, the paper looks essentially descriptive. It would improve its impact by, at least, discussing the physiological and pathological meaning of these profound differences in barrier functions in the different regions of the brain.

*Reviewer #2:*

In this manuscript, Wang et al., address a critical question regarding the mechanisms that prevent blood vessels in circumventricular organs (CVO) to acquire blood-brain barrier (BBB) properties compared to the rest of the central nervous system (CNS) blood vessels. The authors attribute the high permeability in CVO blood vessels to the low levels of beta-catenin signaling in these endothelial cells compared to the rest of the CNS. For example, endothelial cells in area postrema is permeable due to high expression of Wnt inhibitory factor-1 (WIF1) in this region. Moreover, overexpression of a stabilized form of beta-catenin in CVO endothelial cells led to a partial induction of BBB properties both structurally and based on the molecular signatures from the RNA seq and ATAC seq data.

Overall, the study is well conceived and executed, and strengthens our understanding of the molecular mechanisms that regulate barrier properties in the CNS endothelial cells versus blood vessels in CVOs as well as the diversity among endothelial cells in distinct CNS regions. However, there are some concerns about the manuscript as follows:

1) A major functional feature of the BBB is to exclude various circulating molecules from entering the CNS. However, a functional assessment of barrier properties in CVO blood vessels after beta-catenin stabilization is missing throughout the paper. The authors need to inject intravenously small or large MW tracers and assess their permeability into the CVOs under conditions of Wnt/beta-catenin activation in endothelial cells.

2) The transcriptional analysis shown in Figures 6, 7, Figure 7—figure supplement 1, Figure 7—figure supplement 2 and Supplementary files 1, 2 clearly demonstrates a partial induction of BBB-gene expression profile in CVO blood vessels. In my opinion, I do not think that this is only due to an incomplete recombinations which is obvious in the variability of the induction of Wnt downstream target genes (*Axin2, Apcdd1* etc.) (Figure 6). Several key transcriptional regulators of the BBB (Foxf2, Foxq1, *Foxl2*, Ppard, Zic3, Sox17 (Hupe et al., 2016) are completely missing in CVO blood vessels after beta-catenin overexpression after examining carefully Supplementary file 1. There seems to be a complete absence of induction of these genes which likely reflects permanent changes in chromatin accessibility. This has not been discussed or addressed in detail in the paper. The data are there, but they are buried in supplementary files. This is a critical point that needs to be highlighted in greater detail.

3) In Figure 5, it would be important to show TEM images of the structural changes in ciliary body blood vessels following beta-catenin stabilization in this organ.

4) Do the blood vessels in CVO acquire Occludin protein expression at cell junctions with beta-catenin overexpression? The transcriptional analysis suggests that they do, but the authors need to show some staining with Occludin antibodies.

5) Figures 7 and 8 need to be better structured to illustrate more clearly the partial shift in gene expression in AP and PP blood vessels after beta-catenin activation. Perhaps some of the key BBB genes that are induced or nor induced in CVO blood vessels can be annotated in the numerous scatter plots. The salient points of these figures are difficult to follow without some proper annotation of key BBB genes or pituitary genes. Additionally, the authors may consider replacing some of the scatter plots with Venn Diagrams illustrating changes in gene expression.

*Reviewer #3:*

The paper by Wang et al. shows, in a nutshell, that inhibition of Wnt-signaling is an important determinant in the maintenance of the fenestrated and permeable features of specific regional vascular beds within or in close contact with the CNS, i.e. the CVOs, choroid plexus, choriocapillaris and the ciliary body vasculature, which are characterized by a non-barrier phenotype, in marked contrast to the blood-brain/retina barrier (BBB) phenotype of the other CNS vasculature. The authors use a floxed *Ctnnb1*-exon3 allele to induce stabilization of beta-catenin in endothelial cells and knockouts for the Wnt inhibitors *Wif1* and *Sostdc1* to show loss of permeability and gain of a BBB-like phenotype in the non-barrier vasculature.

All in all this is a beautiful study that deserves publication without additional experimental work. While the mechanistic insights are valuable, they don't come as a surprise given prior literature, in part from the Nathan's lab, demonstrating the importance of Wnt signaling for induction and maintenance of the BBB. What is equally valuable to the field, and what impressed me a lot during the reading was the beautiful morphological analyses and the comprehensive and scholarly overview of the field and its history. The manuscript is clearly written and partly a wonderful read. Perhaps the RNA seq and ATAC Seq section in the Results could be condensed and somewhat simplified to enhance the reading. The conclusions of this part are straightforward and simple and the data description would therefore benefit from some shortening.

I have a few suggestions for additional citations:

The paper from Phoenix et al. (Gilbertson lab) deserves mentioning, as it beautifully illustrates the power of Wnt-inhibition to reprogram the BBB into a fenestrated non-barrier vasculature: DOI:https://doi.org/10.1016/j.ccell.2016.03.002

In conjunction with the discussion of MFSD2A, I think David Silver's ground-breaking work on the lipid-transporting capacity of this molecule should be mentioned along with the reference to Ben-Zvi et al. (Gu lab): doi: 10.1038/nature13241

The similarity between the choroid plexus and ciliary body is striking and could be additionally cited by e.g. the work from Maria Lehtinen's lab on tumor development in these structures: doi: 10.1016/j.ajpath.2018.02.009.

---

## [Author Response]

Summary and essential revisions:Reviewers agreed that the study is interesting and the major conclusions of the paper, once substantiated by additional data as outlined in the reviews, would in principle be sufficient to warrant publication in eLife. The two major aspects to revise are:1) Better characterization of the permeability properties of the CVO endothelial cells;

We have performed a permeability analysis using Sulfo-NHS-biotin and the results are shown in panels H and I in the newly expanded Figure 5. Stabilizing beta-catenin in endothelial cells (ECs) results in dramatically reduced vascular permeability in the CVOs and modestly reduced permeability in the choroid plexus. Thank you for suggesting this experiment.

2) A more precise characterization of the composition of endothelial cell junctions in the different conditions.

We have expanded Figure 3 and added three new supplementary figures showing the induction of ZO1 and Occludin in CVO endothelial cells in response to beta-catenin stabilization. These figures also show the induction of the multidrug resistance transporter MDR1 in CVO endothelial cells in response to beta-catenin stabilization. Multiple figures show induction of CLDN5 in CVO ECs in response to beta-catenin stabilization.

Reviewer #1:In the present paper Wang et al. study the role of canonical Wnt /beta-catenin signaling on the permeability properties of the capillaries of the circumventricular organs and choriocapillaris. These vessels present different anatomical and functional characteristics that strongly influence their low control of permeability. Applying excellent imaging analysis and a large set of murine strains of genetically modified mice the authors bring evidence that low beta-catenin signaling is responsible, at least to a significant extent, for low permeability control. This observation is intriguing since the same signaling pathway appears to be required for correct differentiation of the blood brain barrier but should be strongly reduced in the CVO vessels to favor low permeability and dynamic exchanges. This work also shows that brain endothelial cells maintain a certain degree of plasticity in the adult being capable to positively or negatively respond to Wnt signalingThe paper is well done and certainly of interest but some aspects should be better defined to make it more complete:1) Along the manuscript the way to define the blood brain barrier is mostly based on upregulation of claudin-5 expression, increase in GLUT-1 and decreased PLVAP. This may be an oversimplification also considering the extreme variety of the vasculature in the brain or CVO capillaries. I strongly recommend addong staining of other tight junction elements such as cingulin or ZOs and possibly of specific transporters, at least in some key figures.

We have added immunostaining for ZO-1, Occludin, and MDR1 in the following new figures or parts of figures: Figure 3E and F, Figure 2—figure supplement 3, Figure 2—figure supplement 4, Figure 3—figure supplement 3. These new data show induction of each of these BBB markers in CVOs in response to beta-catenin stabilization.

2) The analysis of chromatin accessibility and RNAseq confirm the fact that beta-catenin is a major player in the maintenance of permeability control in the different regions of the brain. However, beta-catenin concentration should be tightly controlled to avoid pathological reactions. In CVO, as reported in this paper, optimal level of beta-catenin activation is due to the fine balance between activators and inhibitors such as WIF-1 and Sostdc1. These observations are novel and of interest. However, overall, the paper looks essentially descriptive. It would improve its impact by, at least, discussing the physiological and pathological meaning of these profound differences in barrier functions in the different regions of the brain.

Thank you for that suggestion regarding an expanded discussion of physiologic and pathologic implications. We have added a new section – titled “Clinical Implications” – to the Discussion to encompass some of our thoughts regarding the implications of the present work for understanding pathologic BBB breakdown and for developing ways to reduce BBB function for therapeutic purposes.

Reviewer #2:In this manuscript, Wang et al., address a critical question regarding the mechanisms that prevent blood vessels in circumventricular organs (CVO) to acquire blood-brain barrier (BBB) properties compared to the rest of the central nervous system (CNS) blood vessels. The authors attribute the high permeability in CVO blood vessels to the low levels of beta-catenin signaling in these endothelial cells compared to the rest of the CNS. For example, endothelial cells in area postrema is permeable due to high expression of Wnt inhibitory factor-1 (WIF1) in this region. Moreover, overexpression of a stabilized form of beta-catenin in CVO endothelial cells led to a partial induction of BBB properties both structurally and based on the molecular signatures from the RNA seq and ATAC seq data.Overall, the study is well conceived and executed, and strengthens our understanding of the molecular mechanisms that regulate barrier properties in the CNS endothelial cells versus blood vessels in CVOs as well as the diversity among endothelial cells in distinct CNS regions. However, there are some concerns about the manuscript as follows:1) A major functional feature of the BBB is to exclude various circulating molecules from entering the CNS. However, a functional assessment of barrier properties in CVO blood vessels after beta-catenin stabilization is missing throughout the paper. The authors need to inject intravenously small or large MW tracers and assess their permeability into the CVOs under conditions of Wnt/beta-catenin activation in endothelial cells.

Thank you for suggesting this experiment. We have performed a permeability analysis using Sulfo-NHS-biotin and the results are shown in panels H and I in the newly expanded Figure 5. This experiment shows that stabilizing beta-catenin results in a dramatically reduced vascular permeability in the CVOs and modestly reduced permeability in the choroid plexus.

2) The transcriptional analysis shown in Figures 6, 7, Figure 7—figure supplement 1, Figure 7—figure supplement 2 and Supplementary files 1, 2 clearly demonstrates a partial induction of BBB-gene expression profile in CVO blood vessels. In my opinion, I do not think that this is only due to an incomplete recombinations which is obvious in the variability of the induction of Wnt downstream target genes (Axin2, Apcdd1 etc.) (Figure 6). Several key transcriptional regulators of the BBB (Foxf2, Foxq1, Foxl2, Ppard, Zic3, Sox17 (Hupe et al., 2016) are completely missing in CVO blood vessels after beta-catenin overexpression after examining carefully Supplementary file 1. There seems to be a complete absence of induction of these genes which likely reflects permanent changes in chromatin accessibility. This has not been discussed or addressed in detail in the paper. The data are there, but they are buried in supplementary files. This is a critical point that needs to be highlighted in greater detail.

Thank you for pointing this out. We agree. We have expanded the Results text to discuss more fully the fact that the gene expression changes induced by beta-catenin stabilization represent only a partial recapitulation of the full BBB gene expression program and we have added two supplementary figures (Figure 6—figure supplements 2 and 3). In the Results, we have added the following sentence: “As seen in Figure 6A, Figure 6A—figure supplement 1, and in the figures that follow, a consistent feature of BBB transcript levels and ATAC peak areas in beta-catenin stabilized pituitary ECs is that, even when induced, the majority are substantially lower than their counterparts in cerebellar ECs.” In the Discussion, we have added the following sentence: “The ATAC analyses represents one step in defining this plasticity at the level of the chromatin state, and – like the transcriptome changes that accompany beta-catenin stabilization – they show that changes in accessible chromatin do not fully recapitulate the BBB pattern, implying a requirement for additional signals or an irreversibility to some chromatin features of the high permeability EC gene expression program.”

3) In Figure 5, it would be important to show TEM images of the structural changes in ciliary body blood vessels following beta-catenin stabilization in this organ.

We have these TEM images, but they are not as aesthetically pleasing as the ones from the pituitary, choroid plexus, and choriocapillaris because, in our hands, cardiac perfusion does not efficiently flush out the contents of the ciliary body capillaries. The result is a background haze of serum proteins in the vessel lumen. This does not affect our ability to see and quantify the fenestrae, but it makes for an image that is not aesthetically pleasing.

4) Do the blood vessels in CVO acquire Occludin protein expression at cell junctions with beta-catenin overexpression? The transcriptional analysis suggests that they do, but the authors need to show some staining with Occludin antibodies.

Yes, Occludin and ZO1 are both induced in CVO endothelial cells in response to beta-catenin stabilization and they both cluster at cell junctions, as does Claudin5. These data have been added in a set of new panels and new supplementary figures (Figure 2—figure supplement 4, Figure 3F, and Figure 3—figure supplement 3).

5) Figures 7 and 8 need to be better structured to illustrate more clearly the partial shift in gene expression in AP and PP blood vessels after beta-catenin activation. Perhaps some of the key BBB genes that are induced or nor induced in CVO blood vessels can be annotated in the numerous scatter plots. The salient points of these figures are difficult to follow without some proper annotation of key BBB genes or pituitary genes. Additionally, the authors may consider replacing some of the scatter plots with Venn Diagrams illustrating changes in gene expression.

Thank you for that comment. We have annotated the same set of 7 key BBB genes in each of the scatterplots in Figure 7 so that the reader can follow the abundances of those transcripts across the different comparisons.

Reviewer #3:[…] All in all this is a beautiful study that deserves publication without additional experimental work. While the mechanistic insights are valuable, they don't come as a surprise given prior literature, in part from the Nathan's lab, demonstrating the importance of Wnt signaling for induction and maintenance of the BBB. What is equally valuable to the field, and what impressed me a lot during the reading was the beautiful morphological analyses and the comprehensive and scholarly overview of the field and its history. The manuscript is clearly written and partly a wonderful read. Perhaps the RNA seq and ATAC Seq section in the Results could be condensed and somewhat simplified to enhance the reading. The conclusions of this part are straightforward and simple and the data description would therefore benefit from some shortening.

Thank you for the positive assessment. In considering the description of the RNAseq and ATAC-seq data, we don’t think that we can shorten it without making it difficult for the reader to follow the logic of the analyses and the conclusions. Also, not all readers will be fully familiar with the “omics” literature in the context of vascular endothelial cells. Some extra explanatory text will help make the work accessible to a broader audience. We have broken this up into an additional section with a new heading, to improve the organization.

I have a few suggestions for additional citations:The paper from Phoenix et al. (Gilbertson lab) deserves mentioning, as it beautifully illustrates the power of Wnt-inhibition to reprogram the BBB into a fenestrated non-barrier vasculature: DOI:https://doi.org/10.1016/j.ccell.2016.03.002In conjunction with the discussion of MFSD2A, I think David Silver's ground-breaking work on the lipid-transporting capacity of this molecule should be mentioned along with the reference to Ben-Zvi et al. (Gu lab): doi: 10.1038/nature13241The similarity between the choroid plexus and ciliary body is striking and could be additionally cited by e.g. the work from Maria Lehtinen's lab on tumor development in these structures: doi: 10.1016/j.ajpath.2018.02.009.

Thank you for these excellent suggestions. We have incorporated all three references into the text and we have expanded the Discussion (a new section titled “Clinical Implications”) to discuss, among other things, the Phoenix et al. study of vascular permeability in medulloblastoma.